# CUG initiation and frameshifting enable production of dipeptide repeat proteins from ALS/FTD C9ORF72 transcripts

Ricardos Tabet[1,2], Laure Schaeffer[3], Fernande Freyermuth[1,2], Melanie Jambeau[1,2], Michael Workman[1], Chao-Zong Lee[1], Chun-Chia Lin[1], Jie Jiang[4], Karen Jansen-West[5], Hussein Abou-Hamdan[6], Laurent Désaubry[6], Tania Gendron[5], Leonard Petrucelli[5], Franck Martin [3] & Clotilde Lagier-Tourenne [1,2]

Expansion of $G_4C_2$ repeats in the *C9ORF72* gene is the most prevalent inherited form of amyotrophic lateral sclerosis and frontotemporal dementia. Expanded transcripts undergo repeat-associated non-AUG (RAN) translation producing dipeptide repeat proteins from all reading frames. We determined *cis*-factors and *trans*-factors influencing translation of the human *C9ORF72* transcripts. $G_4C_2$ translation operates through a 5'–3' cap-dependent scanning mechanism, requiring a CUG codon located upstream of the repeats and an initiator Met-tRNA$^{Met}_i$. Production of poly-GA, poly-GP, and poly-GR proteins from the three frames is influenced by mutation of the same CUG start codon supporting a frameshifting mechanism. RAN translation is also regulated by an upstream open reading frame (uORF) present in mis-spliced *C9ORF72* transcripts. Inhibitors of the pre-initiation ribosomal complex and RNA antisense oligonucleotides selectively targeting the 5'-flanking $G_4C_2$ sequence block ribosomal scanning and prevent translation. Finally, we identified an unexpected affinity of expanded transcripts for the ribosomal subunits independently from translation.

[1] Department of Neurology, MassGeneral Institute for Neurodegenerative Disease (MIND), Massachusetts General Hospital and Harvard Medical School, Boston, MA 02114, USA. [2] Broad Institute of Harvard University and MIT, Cambridge, MA 02142, USA. [3] Architecture et Réactivité de l'ARN, UPR 9002, Université de Strasbourg, CNRS, F-67000 Strasbourg, France. [4] Ludwig Institute for Cancer Research, University of California San Diego, La Jolla, CA 92093, USA. [5] Department of Neuroscience, Mayo Clinic, Jacksonville, FL 32224, USA. [6] Therapeutic Innovation Laboratory (UMR 7200), Faculty of Pharmacy, CNRS/University of Strasbourg, 67401 Illkirch, Cedex, France. Correspondence and requests for materials should be addressed to F.M. (email: f.martin@ibmc-cnrs.unistra.fr) or to C.L.-T. (email: clagier-tourenne@mgh.harvard.edu)

Amyotrophic lateral sclerosis (ALS) and frontotemporal dementia (FTD) are devastating neurodegenerative disorders with a considerable clinical and pathological overlap, which is further substantiated by the discovery of C9ORF72 repeat expansions as the most frequent genetic cause for both diseases[1,2]. Indeed, expansion of a $G_4C_2$ hexanucleotide repeat in the first intron of the C9ORF72 gene is identified in ~40% and ~25% of familial ALS and FTD, respectively, as well as 5% of sporadic patients[3]. The number of $G_4C_2$ repeats is normally lower than 30 and can extend to several hundred repeats in patients. As in other microsatellite diseases, C9ORF72 expansions are transcribed from both sense and antisense strands (reviewed in ref. [4]). Bidirectional transcription of the C9ORF72 locus results in the production of transcripts containing either $G_4C_2$ or $C_4G_2$ repeats that accumulate into RNA foci[1,5–10]. The $G_4C_2$-containing RNAs were proposed to form G-quadruplex secondary structures and sequester several RNA-binding proteins (RBPs) including hnRNP H1/F, ALYREF, SRSF2, hnRNPA1, hnRNPA3, ADARB2, Pur-α, and Nucleolin (reviewed in ref. [4]). In addition, C9ORF72 expanded transcripts are translated into dipeptide repeat (DPR) proteins through unconventional translation, known as repeat-associated non-AUG (RAN) translation[11]. RAN translation occurs in absence of an AUG start codon, in multiple reading frames of the same repeat-containing transcript, and within coding as well as non-coding regions[12]. This mechanism has now been described in several microsatellite expansion diseases, including spinocerebellar ataxia type 8 (SCA8)[11], myotonic dystrophy (DM1 and DM2)[11,13], Huntington's disease (HD)[14], fragile X-associated tremor/ataxia syndrome (FXTAS)[15], spinocerebellar ataxia type 31[16], and C9ORF72 ALS/FTD[10,17–20]. Both $G_4C_2$ sense and $C_4G_2$ antisense transcripts are translated from the three coding frames into five DPR proteins, which aggregate in C9ORF72 ALS/FTD patients[10,13,18–21]. Poly-Glycine-Alanine (poly-GA) and poly-Glycine-Arginine (poly-GR) are translated from the sense strand $G_4C_2$ transcripts, while poly-Proline-Alanine (Poly-PA) and poly-Proline-Arginine (poly-PR) are produced from the antisense strand $C_4G_2$ RNA. Poly-Glycine-Proline (poly-GP) may be produced from both RNA strands. These DPR proteins are the main components of cytoplasmic p62-positive, TDP-43-negative aggregates that represent a unique pathological hallmark in C9ORF72 ALS/FTD patients[22,23]. Evidence supports that DPR proteins, in particular arginine-rich poly-GR and poly-PR proteins, are toxic and play a central role in neurodegeneration due to C9ORF72 expansions (reviewed in ref. [24]).

However, how RAN translation of C9ORF72 expanded transcripts occurs and which factors are required is unknown. Translation initiation of canonical mRNAs is a complex process that requires numerous eukaryotic initiation factors (eIFs) and is crucial for regulation of gene expression. The 40S ribosomal subunit binds to the 5′ cap and then scans along the mRNA until encountering an initiation codon. Most of the regulation is exerted at the first stage, where the AUG start codon is identified and decoded by the methionyl-tRNA specialized for initiation (Met-tRNA$^{Met}_i$)[25]. The efficiency of start codon selection is strongly influenced by surrounding sequences and the recruitment of eIFs. Certain viral and cellular messenger RNAs escape the canonical translation pathway to attract the ribosomes in a cap-independent scanning mechanism. These RNAs contain highly structured sequence, called internal ribosome entry site (IRES), mimicking initiation factors to directly recruit the ribosome at the start codon[26,27]. Repeat-containing RNAs may also adopt stable structures, such as stem loops or G-quadruplexes and an IRES-like mechanism could be at the origin of RAN translation in microsatellite expansion diseases[12,28–32]. Against this hypothesis, RAN translation of CGG repeats associated with FXTAS was recently shown to involve a canonical cap-dependent scanning mechanism[33]. The cis-factors and trans-factors influencing the translation of the human C9ORF72 expansion transcripts are not yet identified. Determining whether hexanucleotide $G_4C_2$ transcripts recruit the ribosome following the canonical translation initiation or using an IRES mechanism is a crucial step for the development of therapeutic approaches targeting RAN translation in C9ORF72 ALS/FTD patients.

Herein, we provide mechanistic insights delineating the different steps needed to recruit the ribosome and initiate RAN translation from $G_4C_2$ expansions to produce poly-GA, GP, and GR proteins. Similar to a canonical mechanism of translation[34], the production of DPR proteins from expanded transcripts requires a 5′cap insertion, involves the initiator methionine and strongly relies on sequences upstream of the repeat. $G_4C_2$ RAN translation proceeds by a 5′–3′ canonical scanning mechanism to start translation at a near-cognate CUG codon and produce DPR proteins by frameshifting. Consistent with this mechanism, we also demonstrate that $G_4C_2$ RAN translation is downregulated by an upstream open reading frame (uORF) present in abnormally spliced C9ORF72 transcripts[35]. Inhibitors of the pre-initiation ribosomal complex and RNA antisense oligonucleotides (ASOs) targeting the sequence upstream of the repeats inhibit $G_4C_2$ RAN translation, confirming a scanning-dependent mechanism that may be targeted for therapeutic intervention. Finally, $G_4C_2$-containing RNAs are found to be associated with ribosomal subunits in a translation independent manner supporting a new RNA gain of function mechanism in C9ORF72 disease.

## Results

**Translation efficiency of $G_4C_2$ RAN translation.** To identify cis-factors and trans-factors influencing the translation of $G_4C_2$ repeats in the context of the C9ORF72 gene, we used a construct containing 66 repeats that was shown to undergo RAN translation in all three frames when expressed in cultured cells and in the mouse central nervous system[20,36]. This construct was modified to generate a series of vectors with different sequences flanking the repeat at the human C9ORF72 locus (Supplementary Fig. 1 and Table 1). Sequences encoding for a specific tag in each of the three reading frames were inserted downstream of the repeat to monitor the production of poly-GA (HA in the +1 frame), poly-GP (His in the +2 frame), and poly-GR (FLAG in the +3 frame). RAN translation from all three reading frames is therefore examined from the same $G_4C_2$ construct.

A cell-free translation assay based on rabbit reticulocyte lysates (RRL) was developed to monitor RAN translation efficiency from C9ORF72 transcripts. In vitro RAN translation was observed in all three frames from capped RNAs with 66 repeats (Fig. 1a–c, Supplementary Fig. 2a). To accurately compare the translation efficiency of the repeat in each frame, we used as reference Renilla luciferases with either HA, His, or FLAG tags under the control of the intergenic region (IGR) IRESs from cricket paralysis virus (CrPV). IRES are structural RNA elements that allow ribosome hi-jacking and trigger translation in a cap-independent manner[26,27]. Among them, IGR promotes highly efficient translation without any AUG start codon, does not need eIF or the initiator tRNA$^{Met}_i$[37,38], and was shown to be efficiently translated in RRL[39]. Indeed, canonical translation under the 5′UTR of the β-globin is only one fold more efficient than translation under the control of the IGR (Supplementary Fig. 2b–d). We compared the efficiency of C9ORF72 RAN translation in the three reading frames with the translation of tagged-luciferase reporter mRNAs that are controlled by the CrPV IGR. A striking difference in translation efficiency was observed between the three frames. Indeed, translation of the capped $(G_4C_2)_{66}$ mRNA in the +1 poly-

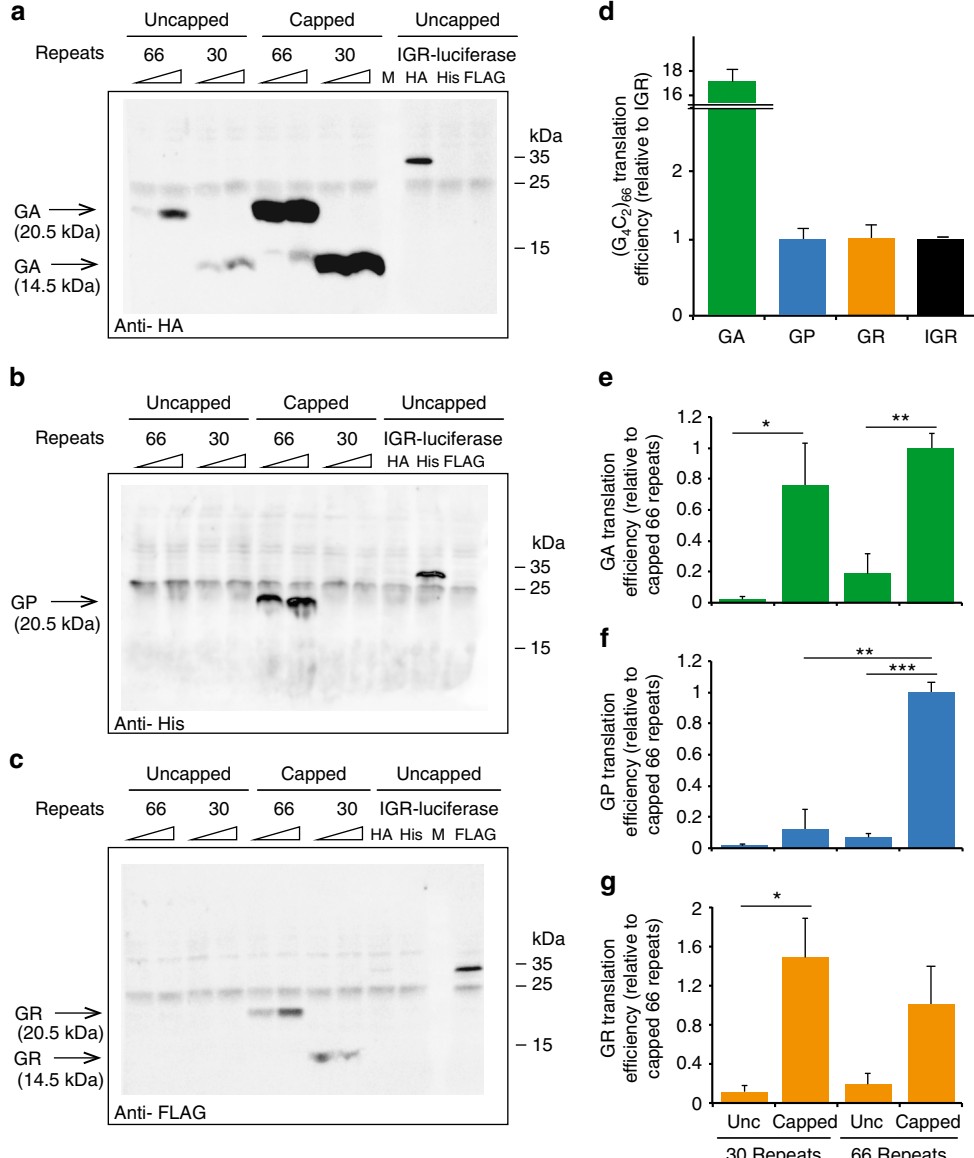

**Fig. 1** $G_4C_2$ RAN translation is length dependent and displays different efficiencies across reading frames. RNA transcripts with $(G_4C_2)_{30}$ or $(G_4C_2)_{66}$ repeats were transcribed in vitro with T7 RNA polymerase, capped or not capped and subjected to translation in rabbit reticulocyte lysate (RRL) system. Increasing RNA concentrations (100 and 200 nM) were used for translation in RRL. RAN translation was probed on immunoblot with antibodies to (**a**) HA tag in the +1 poly-GA frame, (**b**) His tag in the +2 poly-GP frame, and (**c**) FLAG tag in the + 3 poly-GR frame. Schematics of constructs with 30 repeats (#3) and 66 repeats (#4) are shown in Figure S1. (**d**) Efficiencies of RAN translation in the different frames were measured relatively to Renilla Luciferase with the corresponding tags driven by the intergenic region (IGR) IRES from the cricket paralysis virus. The efficiencies of RAN translation from capped RNAs were compared to uncapped RNAs at 100 nM for (**e**) poly-GA, (**f**) poly-GP, (**g**) poly-GR with 30 or 66 $G_4C_2$ repeats, relatively to the capped 66 repeats. Graphs represent mean ± SEM, $n = 3$. Student's $t$-test, $*p \leq 0.05$; $**p \leq 0.01$; $***p \leq 0.001$

GA frame was 17 times more efficient than the IGR-luciferase reporter (Fig. 1a, d). In contrast, translation efficiency from poly-GP in the +2 frame and poly-GR in the +3 frame was equivalent to the translation of IGR-luciferase (Fig. 1b–d). Notably, poly-GA aggregates are the most prevalent DPR proteins accumulated in post-mortem brain samples from *C9ORF72* ALS/FTD patients (Supplementary Fig. 3)[17,40] supporting that translation of the *C9ORF72* repeat is most efficient in the +1 frame both in vitro and in vivo.

We also uncovered that the size of the expansion does not equally influence translation of the different frames. Production of poly-GP in the +2 frame was strongly influenced by the size of the repeat when comparing 30 and 66 repeats (Fig. 1b, f, Supplementary Fig. 1; constructs #3 vs. #4). In contrast, no

significant difference was observed for poly-GA or poly-GR, which were equally expressed from both 30 and 66 $G_4C_2$ repeats (Fig. 1a, c, e, g).

**Cap-dependent $G_4C_2$ translation initiates with methionine.** Our in vitro assay provided the opportunity to determine whether RAN translation of the *C9ORF72* repeat depends on the presence of a 5′m7G cap. Levels of poly-GA produced from 66 repeats increased more than five times when transcripts were capped (Fig. 1a, e) and poly-GP/GR syntheses were strongly repressed in absence of the cap, supporting a canonical cap-dependent mechanism of translation for all three DPR proteins (Fig. 1a, b, c, e, f, g).

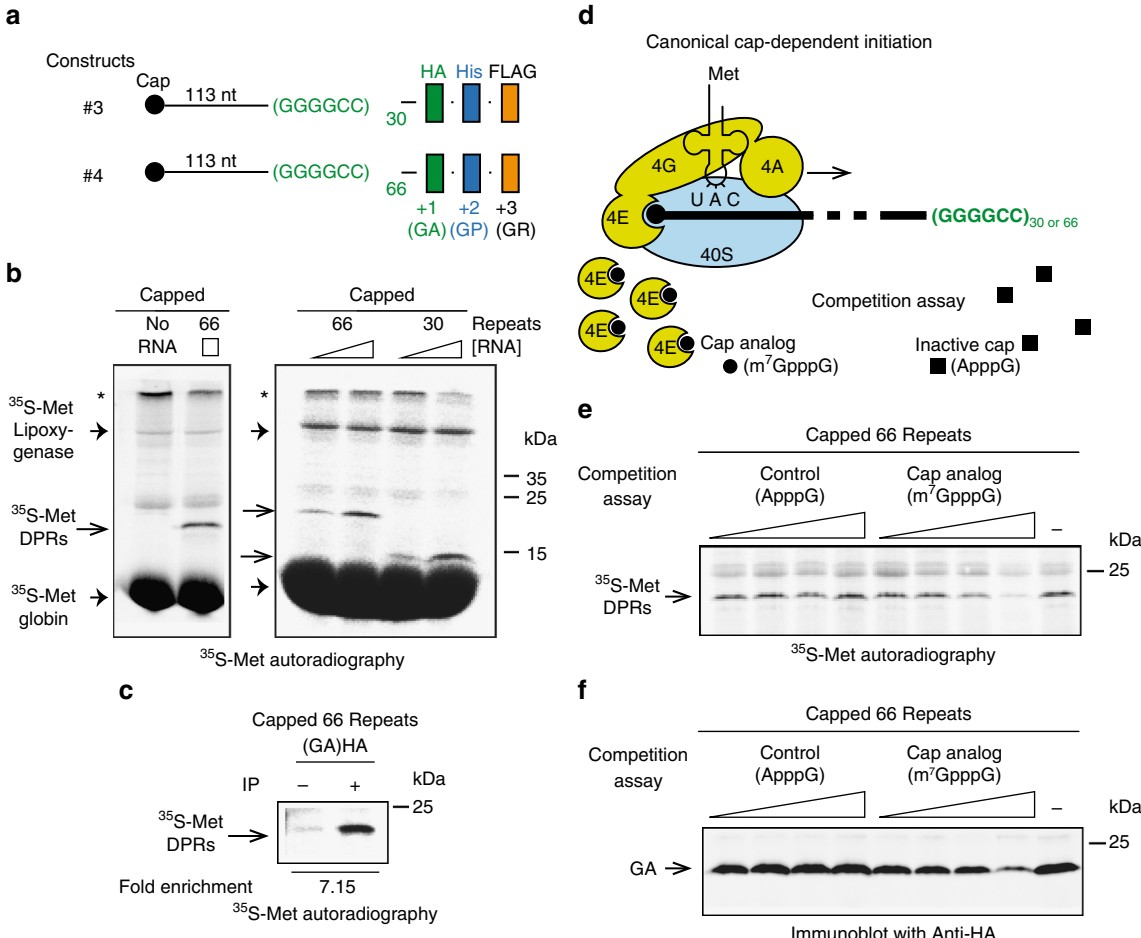

**Fig. 2** $G_4C_2$ RAN translation is cap-dependent and initiates with a methionine. (**a**) Schemes of the RNA with $(G_4C_2)_{30}$ (#3) or $(G_4C_2)_{66}$ (#4) repeats that were transcribed in vitro with T7 RNA polymerase, capped and subjected to translation in RRL. (**b**) Translation was performed in the presence of [$^{35}$S]-methionine and capped RNA #3 or #4 at 100 and 200 nM. RAN translation products were detected by autoradiography. Asterisk indicates bands in the stacking gel. (**c**) Translation was performed in presence of [$^{35}$S]-methionine and capped RNA #4 followed by immunoprecipitation with antibody against HA-tag and detection of immunoprecipitated [$^{35}$S]-methionine proteins by autoradiography. (**d**) Scheme of the canonical translation involving the cap-binding protein eukaryotic initiation factor 4E (eIF4E), the protein platform (eIF4G) and the helicase (eIF4A) that recruit the 40S ribosomal subunit. This pre-initiation complex scans the 5′ of the transcript for an appropriate start codon. Compounds used for the competition assay in (**e**) and (**f**) are represented by dark circles and squares for the cap analog ($m^7$GpppG) and the inactive form (ApppG), respectively. (**e**–**f**) Translation was performed in presence of [$^{35}$S]-methionine, capped $(G_4C_2)_{66}$ RNA #4 and an increased concentration of inactive cap (control, ApppG) or cap analog (competitor of the cap, $m^7$GpppG). [$^{35}$S]-methionine RAN translation products and poly-GA were detected by (**e**) autoradiography and (**f**) immunoblot with an antibody against HA-tag, respectively

Canonical translational initiation consists of base-pairing between the initiator Met-tRNA$^{Met}_i$ anticodon and the AUG start codon. The incorporation of [$^{35}$S]-methionine during the translation of transcripts expressing 30 repeats (#3) or 66 repeats (#4) was measured to determine whether RAN translation requires Met-tRNA$^{Met}_i$ for the production of DPR proteins (Fig. 2a, b). Notably, the sequence of the transcripts #3 and #4 do not contain any AUG codon and the presence of [$^{35}$S]-methionine in RAN products cannot derive from the incorporation of an internal methionine (Supplementary Fig. 1 and Table 1). A specific [$^{35}$S]-methionine band was detected at the expected 14.5 and 20.5 kDa molecular weight from constructs expressing 30 and 66 repeats, respectively (Fig. 2b). The level of [$^{35}$S]-methionine labeled polypeptide(s) was proportional to RNA concentration indicating that RAN translation is observed in sub-saturating conditions. Immunoprecipitation of poly-GA products with a HA-specific antibody confirmed that RAN translation initiates with the incorporation of a methionine residue (Fig. 2c). To further demonstrate that $G_4C_2$ RAN translation starts with a

methionine, we inhibited the activity of the methionylated initiator tRNA$^{Met}$ carrier eIF2 by inducing the phosphorylation of its α subunit with poly(I:C)/salubrinal treatment as previously described[41] (Supplementary Fig. 4a). While this treatment did not have any impact on the non-canonical translation of IGR-renilla luciferase, it inhibited the translation of a capped-dependent renilla luciferase reporter and the incorporation of [$^{35}$S]-methionine in DPR products (Supplementary Fig. 4b).

To gain further insights in the mechanism of RAN translation, we investigated the requirement of eukaryotic initiation factor eIF4E. In canonical translation the cap binding protein eIF4E is part of a larger complex called eIF4F, which also contains the platform protein eIF4G and the RNA helicase eIF4A[34] (Fig. 2d). To test whether eIF4E is involved in the RAN translation of $G_4C_2$ repeats we monitored the translation efficiency in the presence of an excess of cap analog ($m^7$GpppG) or its non-functional ApppG counterpart. The competition assay was performed in RRL (Fig. 2e, f) and wheat germ extract (WGE) (Supplementary Fig. 4c, d), a highly cap-dependent system[42]. Increasing

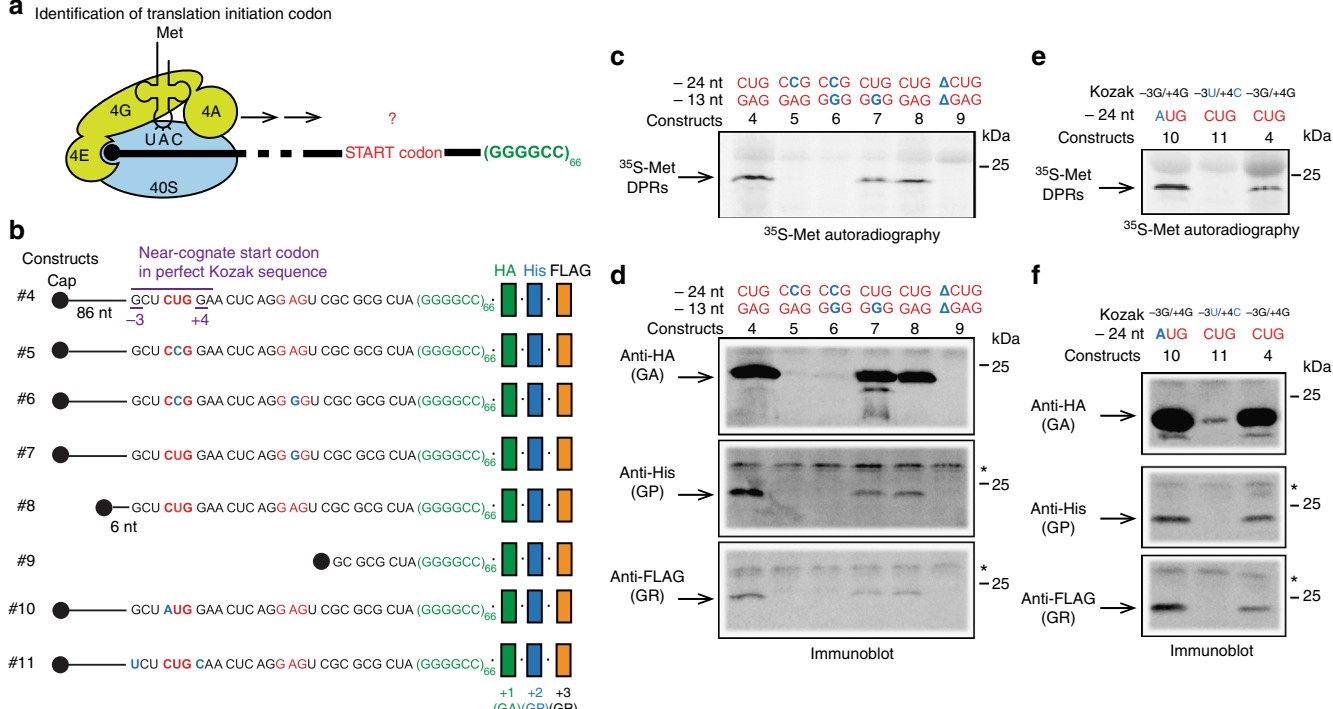

**Fig. 3** $G_4C_2$ RAN translation of all reading frames initiates at the same near-cognate CUG start codon in RRL. (**a**) Scheme of the pre-initiation complex loaded at the 5′cap and the 40S ribosomal subunit ready to scan toward the start codon. RAN translation occurs in absence of an AUG codon. (**b**) Schemes of the transcripts #4 to #11 showing mutations in the 5′ flanking sequence of $(G_4C_2)_{66}$ used in (**c-f**). Construct #4 contains the native sequence of 113 nucleotides upstream of the $G_4C_2$ repeat. Construct #5 has a CUG > CCG mutation (blue nucleotide) in a near-cognate start codon located in a perfect Kozak sequence 24 nucleotides upstream of the repeat. Construct #6 has CUG > CCG and GAG > GGG (blue nucleotides) mutations in two potential start codons located 24 and 13 nucleotides upstream of the repeat, respectively. Construct #7 contains a GAG > GGG mutation in a potential near-cognate start codon located 13 nucleotides upstream of the repeat. Constructs #8 and #9 harbor a deletion leaving 33 nucleotides (including CUG and GAG codons) and eight nucleotides (deleting both CUG and GAG codons) upstream of the repeat, respectively. Construct #10 has a CUG > AUG mutation in the near-cognate start codon. Construct #11 has GCUCUGG > UCUCUGC mutations in the Kozak sequence. (**c-f**) Translation was performed in presence of [$^{35}$S]-methionine using each RNA variant separately (#4 to #11). (**c** and **e**) [$^{35}$S]-methionine RAN translation products were detected by autoradiography. (**d** and **f**) Poly-GA, poly-GP, and poly-GR were detected by immunoblot using antibodies against HA-tag, His-tag, and FLAG-tag, respectively. Asterisk indicates unspecific proteins translated in the RRL system

concentrations of cap analog, but not ApppG, lead to eIF4E titration thereby affecting the efficiency of eIF4F-dependent translation. The levels of [$^{35}$S]-methionine-DPRs (Fig. 2e, Supplementary Fig. 4c) and poly-GA accumulation (Fig. 2f, Supplementary Fig. 4d) were reduced by increased concentrations of cap analog, demonstrating the role of the canonical initiation factor eIF4E in *C9ORF72* RAN translation.

**$G_4C_2$ translation initiates at a near-cognate CUG start codon.** We next sought to identify the codon(s) used to initiate translation of *C9ORF72* transcripts (Fig. 3a). The presence of a single band on SDS-PAGE for the different DPR products (Fig. 1), corroborated by [$^{35}$S]-methionine labeling (Fig. 2), suggests that the translation of $G_4C_2$ starts at a specific position. In addition, in vitro RAN translation products obtained from 66 repeats had the same estimated molecular weight of 20.5 kDa in all three frames (Fig. 1a–c) suggesting that translation in the different frames is initiated from a single or neighboring start codons.

A candidate start site is a near-cognate CUG codon located 24 nucleotides upstream of the repeats in the +1 frame and embedded in a perfect Kozak sequence[43] (G/A in −3 and G in +4) (Supplementary Fig. 1, Fig. 3b, and Supplementary Table 1). Site-directed mutagenesis of this codon from CUG to CCG was sufficient to abolish the production of [$^{35}$S]-methionine labeled DPR proteins in RRL, demonstrating that this CUG is used as

start codon to translate *C9ORF72* $G_4C_2$ repeats (Fig. 3b, c; construct #4 vs. #5). In contrast, a point mutation from GAG to GGG in another putative start site located 13 nucleotides upstream of the repeats in the +2 frame only slightly reduced the level of [$^{35}$S]-methionine DPR proteins (Fig. 3b, c; construct #4 vs. #7). Transcripts containing mutations at both putative start codons confirmed the necessity of the CUG codon to initiate RAN translation of the *C9ORF72* repeat (Fig. 3b, c; construct #4 vs. #6). This was further corroborated by using constructs with 5′ truncations either preserving the near cognate CUG codon (#8) or deleting the entire region (#9) (Fig. 3b, c; construct #8 vs. #9). Importantly, immunoblot analyses revealed that syntheses of all three DPRs, poly-GA, poly-GP, and poly-GR, are equally disabled by the CUG mutation located 24 nucleotides upstream of the repeats (Fig. 3d; construct #4 vs. #5). In contrast, mutation of the GAG codon located 13 nucleotides upstream of the repeats reduced the levels of the three DPRs without abolishing their production (Fig. 3d; construct #4 vs. #7). Site-directed mutagenesis of the near cognate CUG codon to a canonical start codon AUG increases the incorporation of [$^{35}$S]-methionine in DPR products (Fig. 3b, e; construct #10 vs. #4) and concomitantly the level of DPRs from all three frames, poly-GA, poly-GP, and poly-GR (Fig. 3b, f; construct #10 vs. #4). Interestingly, mutating the Kozak sequence inhibits the production of DPR proteins detected by [$^{35}$S]-methionine autoradiography (Fig. 3b, e; construct #11 vs.

#4), as well as immunoblots for poly-GA, poly-GP, and poly-GR (Fig. 3b, f; construct #11 vs. #4). This striking result demonstrates that RAN translation producing DPR proteins from the three frames starts at the same CUG codon, and implies that production of poly-GP and poly-GR requires frameshifting events, +1 and −1, respectively. An additional smaller poly-GA product was translated from construct #7 suggesting that mutation of G<u>A</u>G to G<u>G</u>G induces another translation initiation event further downstream in frame +1 that is less efficient than initiation at CUG. The frameshifting necessary to produce +2 (poly-GP) and +3 (poly-GR) DPR proteins might explain the yield of DPR productions observed in Fig. 1 and patient tissues (Supplementary Fig. 3). Indeed, poly-GA translated from the +1 frame is the predominant DPR protein, poly-GP and poly-GR require one frameshifting event (−1 or + 1) and are therefore significantly less produced.

The crucial role of the CUG translation initiation codon located 24 nucleotides upstream of the *C9ORF72* repeat was further confirmed in vivo by expressing 66 repeats with either a CUG codon (construct #4) or its mutated CCG version (construct #5) in human neural progenitor cells (ReNcell VM)[44], mouse motor neuron-like cells (NSC-34), and human embryonic kidney 293T cells (HEK293T) (Fig. 4). Immunoblots using antibodies that recognize each DPR protein identified products at a comparable molecular weight in the three cell types and RRL demonstrating similar RAN translation of the wild-type construct (#4) in all systems (Figs. 3 and 4). RAN translation of poly-GA and poly-GP was abolished by mutation of the CUG codon in human neural progenitors (Fig. 4b, c, Supplementary Fig. 5a, b; construct #4 vs. #5) and motor neuron-like cells (Fig. 4d, e; construct #4 vs. #5), confirming results observed in RRL (Fig. 3; construct #4 vs. #5). RAN translation of poly-GR could not be detected with any of the constructs in these cell lines. As shown in RRL experiments (Fig. 3), $G_4C_2$ RAN translation in the poly-GA +1 frame and the poly-GR +3 frame was also abolished by mutation of the CUG codon in HEK293T cells (Fig. 4f, g, Supplementary Fig. 5d; construct #4 vs. #5). However, in contrast to RRL and the two neuronal models, mutating the CUG codon did not inhibit production of poly-GP in HEK293T cells but instead induced a 20% increase detected by antibodies recognizing either poly-GP (Fig. 4f, g; Supplementary Fig. 5e; construct #4 vs. #5) or the HIS tag (Supplementary Fig. 5c). This observation supports that poly-GP translation from an alternative start site may be influenced in HEK293T by additional *trans*-acting factor(s) that are absent in RRL, motor neuron-like NSC-34 and neural progenitor cells. Overall, these results identify a mechanism where cap-dependent translation of the *C9ORF72* $G_4C_2$ repeat requires Met-tRNA$^{Met}_i$ to initiate translation in all reading frames at a near-cognate CUG codon located upstream of the expansion.

**An uORF represses $G_4C_2$ translation**. Recently, Niblock et al. identified poly-adenylated *C9ORF72* RNA species that retain the repeat-containing intron 1 and in which downstream exons are correctly spliced[35]. This finding opens the possibility that $G_4C_2$ RAN translation occurs from a *C9ORF72* mRNA variant with an enlarged 5′-untranslated region containing the $G_4C_2$ repeats. Notably, retention of intron 1 creates a potential uORF with 55 codons flanked by an AUG start codon and two consecutive stop codons (UGA and UAA) (Supplementary Fig. 1 and Table 1). Emerging evidence suggests that the presence of uORF may regulate the expression of downstream ORF[25,45,46]. Indeed, translation of uORFs located in the 5′UTRs of transcripts often inhibits translation of the downstream ORF likely by reducing its accessibility to the preinitiation complex[47,48]. Hence, we tested whether the uORF created by the retention of intron 1 in

*C9ORF72* transcripts may influence RAN translation of DPR proteins (Fig. 5a). We generated a construct with 66 repeats and the entire 5′ end sequence of *C9ORF72* starting with exon 1A (Fig. 5b, Supplementary Fig. 1; construct #1). The uORF was found to strongly repress RAN translation in all frames of *C9ORF72* repeat. Indeed, [$^{35}$S]-methionine labeled DPR proteins were not detected in presence of the uORF (Fig. 5c, construct #1 vs. #4). Immunoblot analysis confirmed the influence of the uORF with a severe reduction of poly-GA (+1 frame) levels and non-detectable poly-GP (+2) and poly-GR (+3) products (Fig. 5d; construct #1 vs. #4). Mutation of the uORF A<u>U</u>G start codon into C<u>G</u>G (Fig. 5b, construct #2) restored $G_4C_2$ RAN translation from all reading frames confirming its role in repressing RAN translation (Fig. 5c, d; construct #2 vs. #1). Overall, these findings strongly support that RAN translation operates through a 5′−3′ scanning mechanism and is regulated by an uORF in *C9ORF72* transcripts that retain intron 1.

**5′−3′ scanning-dependent mechanism for $G_4C_2$ translation**. To further demonstrate that RAN translation uses a canonical 5′−3′ scanning mechanism we investigated whether the eIF4A, an RNA helicase required for efficient scanning during translation initiation, is involved in $G_4C_2$ RAN translation (Fig. 6a−e). FL3, an eIF4A-specific inhibitor[49], was found to reduce RAN translation in RRL as demonstrated by the levels of [$^{35}$S]-methionine labeled DPR proteins (Supplementary Fig. 6a,b; construct #4) and poly-GA (Fig. 6b, Supplementary Fig. 6c; construct #4) generated from two different concentrations of expanded RNAs. Consistently, FL3 treatment significantly reduced the levels of poly-GA, poly-GP, and poly-GR in HEK293T (Fig. 6c, d) without affecting the level of the repeat-containing transcripts (Fig. 6e). To confirm the role of eIFs and a 5′−3′scanning mechanism in RAN translation, we used a longer transcript that includes exon 1a and the entire intronic region upstream of the *C9ORF72* repeat with a AUG > C<u>G</u>G mutation in the uORF start codon (Supplementary Fig. 6a; construct #2). Consistent with our previous results, production of DPR proteins was partially restored in presence of the mutated uORF, but was strongly inhibited after treatment with FL3 (Supplementary Fig. 6a−c; construct #2). Another important component for scanning is the platform eIF4G, which links the cap binding factor eIF4E with the small ribosomal subunit (Fig. 6a). To investigate whether eIF4G is required for $G_4C_2$ RAN translation we used 4EIRCat, an inhibitor that prevents the direct interaction between eIF4E and eIF4G[50]. Consistently, synthesis of poly-GA from two different RNA concentrations was also reduced by 4EIRCat (Fig. 6b). Finally, we found that both edeine and cycloheximide completely inhibited the RAN translation from all three reading frames (Fig. 6b, Supplementary Fig. 6b−d). Edeine is a translation inhibitor that prevents the interaction of Met-tRNA$^{Met}_i$ anticodon with the start codon in the P site of the ribosome (Fig. 6a). Cycloheximide binds between the E site and P site of the ribosome and thereby blocks translocation to the next codon (Fig. 6a)[51]. The profound effect of these inhibitors on RAN translation is consistent with our previous results showing that $G_4C_2$ RAN translation uses a canonical translation mechanism and initiates at a CUG codon with Met-tRNA$^{Met}_i$ anticodon interaction in the P site of the ribosome (Figs. 2−4). Overall, the effect of specific translation inhibitors on the production of DPR proteins demonstrate that $G_4C_2$ RAN translation requires eIF4F components (4E, 4G and 4A) to promote efficient cap-dependent 5′−3′ scanning.

**Inhibition of $G_4C_2$ translation by RNA ASOs**. We previously showed that DNA ASOs targeting sense strand $G_4C_2$-containing transcripts mediate their cleavage through action of the primarily

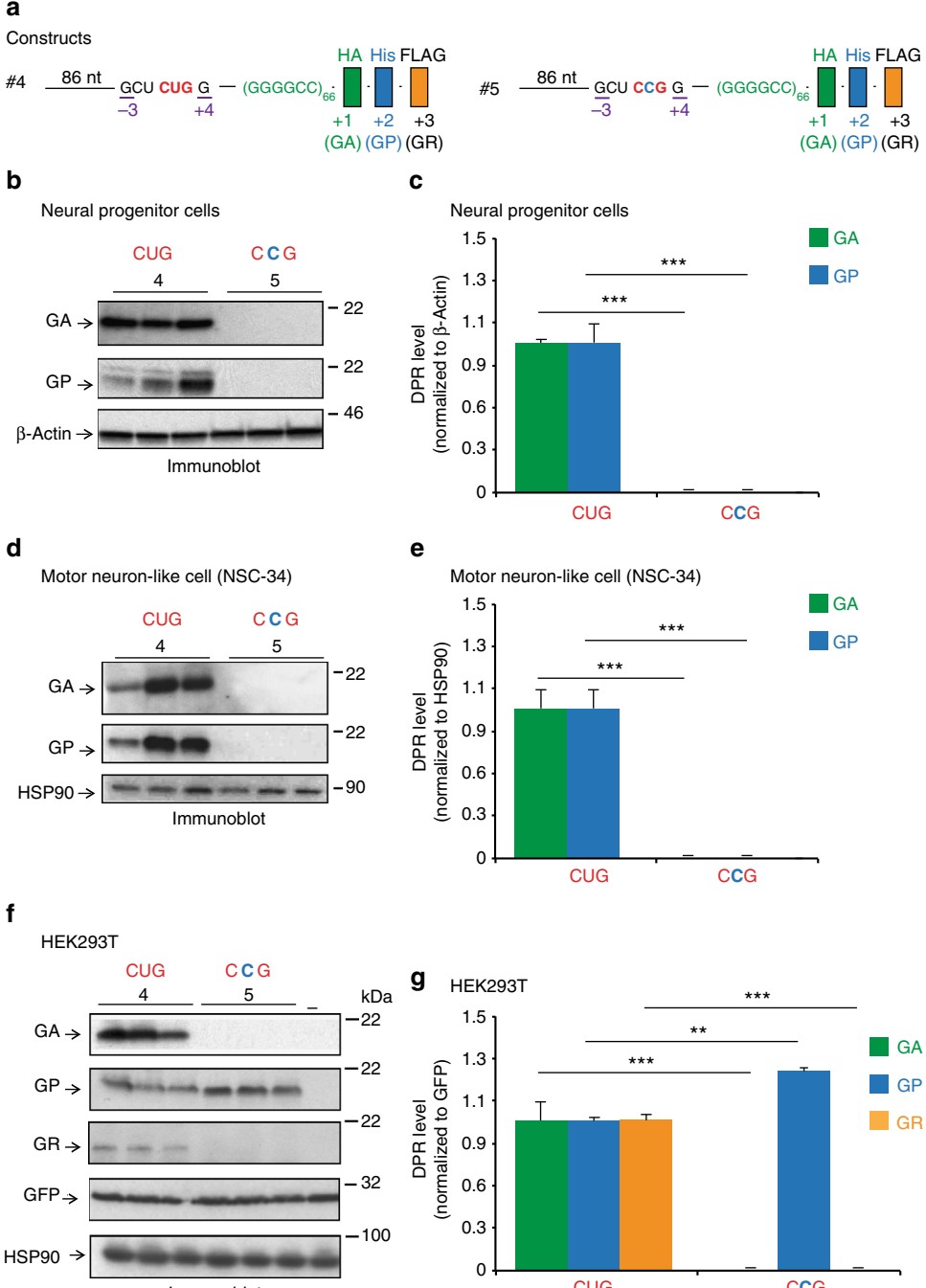

**Fig. 4** Poly-GA, poly-GP, and poly-GR RAN translation initiate at the near-cognate CUG start codon in cells. (**a**) Schematic representations of constructs #4 and #5 containing the near cognate start codon CUG or mutant CCG upstream of $(G_4C_2)_{66}$ repeats. These constructs are driven by a CMV early enhancer/chicken β actin (CAG) promoter. Human neural progenitor cells (**b–c**), mouse motor neuron like cells (NSC-34) (**d–e**) and human HEK293T cells (**f–g**) were co-transfected with the constructs #4 or #5 along with a GFP plasmid reporter. GFP, Hsp90 or β-Actin proteins were analyzed by immunoblot to control for the transfection efficiency and protein loading. Poly-GA, poly-GP, and poly-GR proteins were identified by immunoblot using antibodies raised against poly-GA, poly-GP, and poly-GR. Levels of the different DPR proteins were quantified and normalized to GFP, HSP90 or β-Actin. Error bars indicate SEM of three independent transfections. Student's *t*- test, ** and *** indicate $p < 0.01$ and $p < 0.001$, respectively

nuclear enzyme RNase H, reducing the level of RNA foci and DPR proteins in a *C9ORF72* transgenic mouse model and patient fibroblasts[7,52]. To determine whether RNA ASOs targeting the 5′ flanking $G_4C_2$ sequence can block the scanning of ribosomes and inhibit RAN translation without inducing RNAse-H-dependent degradation, we generated ASOs selectively targeting the region upstream of the repeats and tested their potency in inhibiting $G_4C_2$ RAN translation in RRL system (Fig. 6f–h). One RNA

*C9ORF72* ASO (RNA-ASO1) was complementary to a sequence that overlaps the near-cognate CUG codon, and two ASOs (RNA-ASO2, RNA-ASO3) were chosen to cover sequences located at 41 and 82 nucleotides distal from the repeats, respectively (Fig. 6f). Corresponding RNA sense oligonucleotides (RNA-SOs) were used as controls (RNA-SO1, RNA-SO2, and RNA-SO3, Fig. 6f). All three RNA-ASOs induced a dose-dependent reduction of DPR proteins produced from the capped $G_4C_2$ 66 repeats RNAs

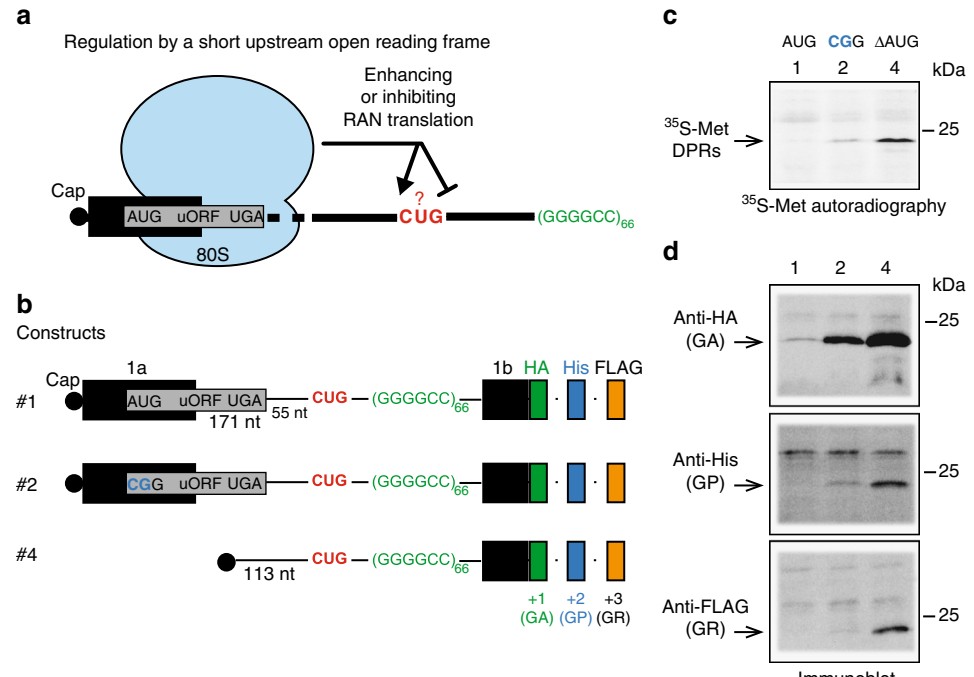

**Fig. 5** RAN translation of $G_4C_2$ repeats is down-regulated by a short upstream open reading frame (uORF). (**a**) Retention of intron 1 in *C9ORF72* repeat-containing transcripts creates an uORF located 226 nucleotides upstream of the start CUG codon. This uORF may inhibit or enhance $G_4C_2$ RAN translation. (**b**) To interrogate the regulation of RAN translation by this uORF, RNAs harboring the 5′ full-length sequence including *C9ORF72* exon 1A (#1) and a AUG > CCG mutation in the uORF start codon (#2) were compared to RAN translation from RNA without the uORF (#4). Black boxes represent exons 1a and 1b and the gray box represents the uORF overlapping exon 1a and intron 1. (**c**) Translation in RRL system was performed in presence of [35S]-methionine and capped RNA (#1, #2, or #4) followed by detection of [35S]-methionine proteins by autoradiography. (**d**) Poly-GA, poly-GP, and poly-GR were detected by immunoblots using antibodies against HA-tag, His-tag, and FLAG-tag, respectively

as measured by [35S]-methionine-labeling (Fig. 6g) and immunoblot (Fig. 6h). In contrast, SO controls did not affect the levels of DPR proteins. These results demonstrate that RNA ASOs targeting the 5′ flanking $G_4C_2$ sequence are sufficient to block RAN translation independently of *C9ORF72* RNAs degradation and identify the 5′–3′ scanning of ribosomes as a potential therapeutic target in *C9ORF72* ALS/FTD.

**$G_4C_2$ RNAs bind ribosomes independently from translation**. To assess ribosome loading onto $(G_4C_2)_{exp}$ RNAs, we performed sucrose gradient analysis with radiolabeled capped RNAs containing either 30 or 66 repeats. As a control for canonical translation we used radiolabelled capped human β-globin mRNA. Radiolabeled capped RNAs with 66 antisense $C_4G_2$ repeats were also used as control for RAN translation (Fig. 7a). Sucrose gradient analysis with 30 and 66 $G_4C_2$ repeat transcripts showed that RNA-containing repeats are mainly associated with polysomes (Fig. 7a, b green graph, Supplementary Fig. 6e orange graph; heavy fractions 0–20). Only a small proportion of RNAs was free (RNP; ribonucleoproteins), associated with the ribosomal small subunit in complex with initiation factors (48S) or with monosomes (80S), which is consistent with active RAN translation (Fig. 1). Since transcripts containing expanded repeats, including $G_4C_2$ RNAs, were recently shown to undergo abnormal phase transition and form gel-like structures in vitro[53], we determined whether the presence of radiolabeled $G_4C_2$ RNAs in the heavy fractions could be due to self-aggregation rather than association with polyribosomes. Against this possibility, $G_4C_2$-free RNAs remained in the light fractions of sucrose gradients strongly supporting that expanded RNAs associate with polyribosomes in RRL. Contrary to the sense $(G_4C_2)_{66}$ RNAs, transcripts

containing the antisense $(C_4G_2)_{66}$ repeat sedimented mainly in the light fractions or were associated to monosomes, consistent with a low translation efficiency of the antisense transcripts (Fig. 7a, b; blue graph)[40]. Unexpectedly, treatment with edeine, that blocks the translation (Fig. 6b) and lead to the accumulation of β-globin mRNA in the light fractions (Fig. 7c, Supplementary Fig. 6f; light fractions 20–40, red graphs), did not prevent loading of polysomes on transcripts with 66 or 30 $G_4C_2$ repeats (Fig. 7c, Supplementary Fig. 6f; heavy fractions 0–20, green and orange graphs). The same abnormal sedimentation of $G_4C_2$ transcripts in heavy fractions was observed after treatment with GMP-PNP, a non-hydrolysable GTP analog that normally leads to the accumulation of the transcripts in the fraction corresponding to the 48S particles, showing that $G_4C_2$ RNAs can recruit ribosomes in a translation-independent manner (Supplementary Fig. 6g). As expected, blocking ribosomal translocation with cycloheximide induced the accumulation of the control β-globin mRNAs in the fraction corresponding to monosomes 80S that are prevented from translocating after assembly (Fig. 7d, Supplementary Fig. 6h; red graphs). In contrast, inhibiting RAN translation with cycloheximide (Fig. 6b, Supplementary Fig. 6b–d) did not prevent ribosomal loading on expanded transcripts with 30 or 66 repeats (Fig. 7d, Supplementary Fig. 6h; heavy fractions 0–20, green and orange graphs). As expected the 80S peak was slightly increased consistent with a small proportion of expanded $G_4C_2$ RNAs being associated with monosomes after cycloheximide treatment, but most transcripts remained present in the heavy fractions despite cycloheximide blockage of translation. Notably, radiolabeled $(G_4C_2)_{66}$ transcripts were more abundant in heavy fractions when they were folded in presence of $K^+$ ions that stabilize G-quadruplex structures, comparatively to $Na^+$ and $Li^+$ ions (Supplementary Fig. 6i). Finally, to confirm that $G_4C_2$ RNAs recruit

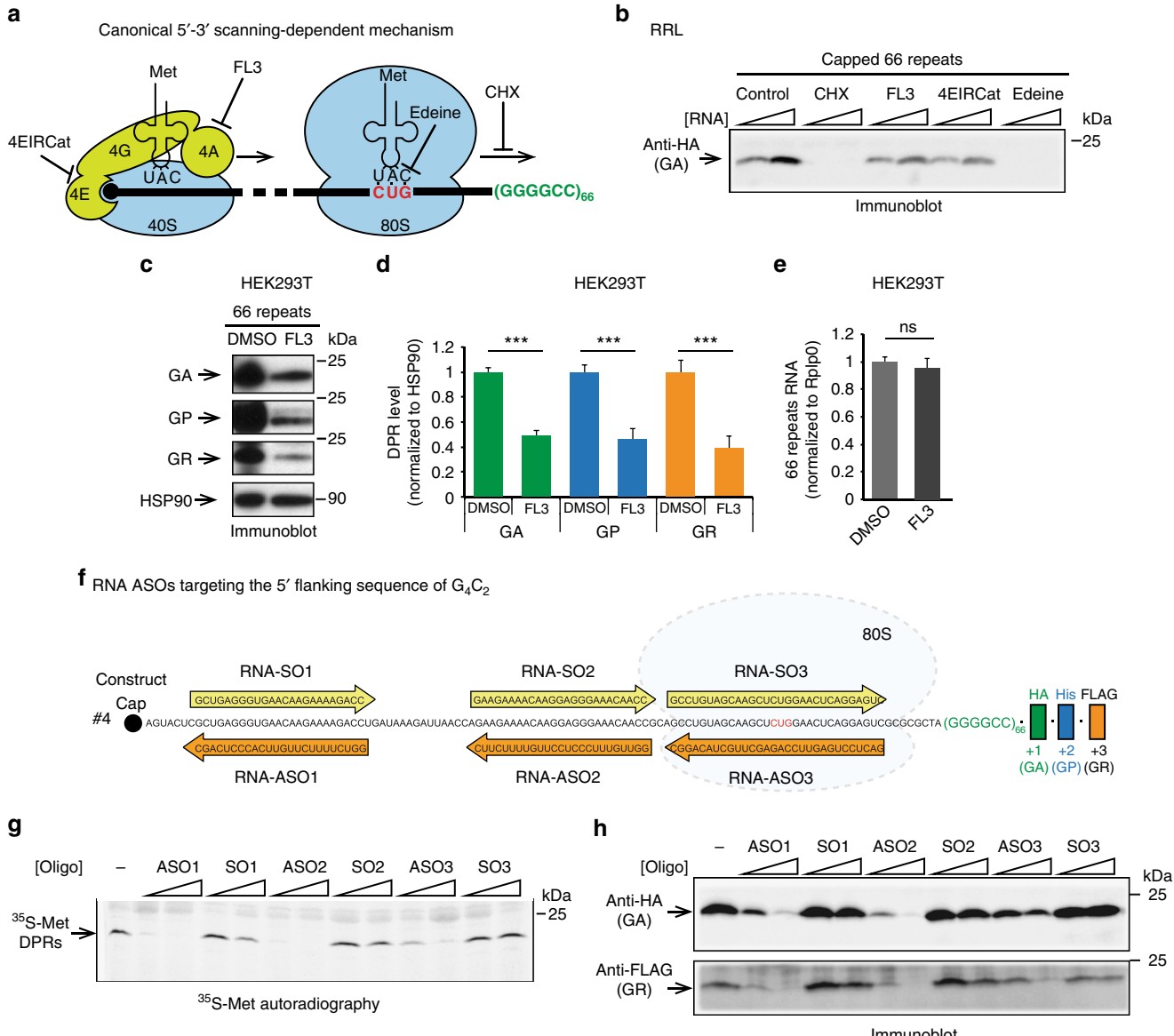

**Fig. 6** Inhibition of RAN translation by eIFs inhibitors and RNA ASOs support a 5′–3′ scanning-dependent mechanism. (**a**) Illustration of translation inhibitors used to delineate the recruitment of the ribosome at the CUG start codon: 4EIRCat prevents the interaction between eIF4E (4E) and eIF4G (4G). FL3 inhibits RNA helicase eIF4A (4A). Edeine blocks the codon–anticodon interaction. Cycloheximide (CHX) blocks the translational elongation. (**b**) Translation was performed in presence of CHX, FL3, 4EIRCat, or Edeine in RRL followed by immunoblot detection of anti-HA (poly-GA) antibody. (**c–e**) HEK293T cells were transfected with the construct #4 expressing 66 $G_4C_2$ repeats and treated with FL3 or DMSO control. (**c**) Immunoblots using antibodies against poly-GA, poly-GP, poly-GR, and HSP-90 proteins. (**d**) Levels of poly-GA, poly-GP, and poly-GR after FL3 treatment were quantified and normalized to HSP90 and DMSO-treated cells. Graphs represent mean ± SEM, $n = 5$. Student's $t$-test, *** indicate $p < 0.001$. (**e**) Levels of repeat-containing transcripts determined by quantitative RT-PCR and normalized to the Rplp0 transcripts and DMSO treated cells. (**f**) Schematic representations of construct #4 with sequences of sense (RNA-SO) and antisense (RNA-ASO) RNA oligonucleotides used to inhibit RAN translation. (**g–h**) Translation of capped $(G_4C_2)_{66}$ RNAs (construct #4) was performed in RRL in presence of two concentrations of sense or antisense RNA oligonucleotides. (**g**) [$^{35}$S]-methionine RAN translation products were detected by autoradiography. (**h**) Poly-GA and poly-GR were detected by immunoblot using anti-HA (Poly-GA) and -FLAG (Poly-GR) antibodies, respectively

the ribosome independently from DPR translation, we performed sucrose gradient analysis with purified ribosomal 40S and 60S. Expanded transcripts with 30 repeats were able to recruit and load several 40S and 60S ribosomal subunits without the need of 5′-cap and any other initiation factors (Fig. 7e). Overall, we demonstrate here that $G_4C_2$ repeat-containing transcripts associate with ribosomal subunits independently of translational factors.

## Discussion

$G_4C_2$ hexanucleotide expansions in the *C9ORF72* gene were recently discovered as the major genetic cause of ALS and FTD, two fatal neurodegenerative disorders[1,2]. Emerging evidence supports pathogenic RNA gain-of-function mechanisms, where expanded $G_4C_2$ transcripts form RNA foci sequestering RNA-binding proteins in the nuclei or undergo RAN translation to produce toxic DPR proteins[4]. We developed robust assays to

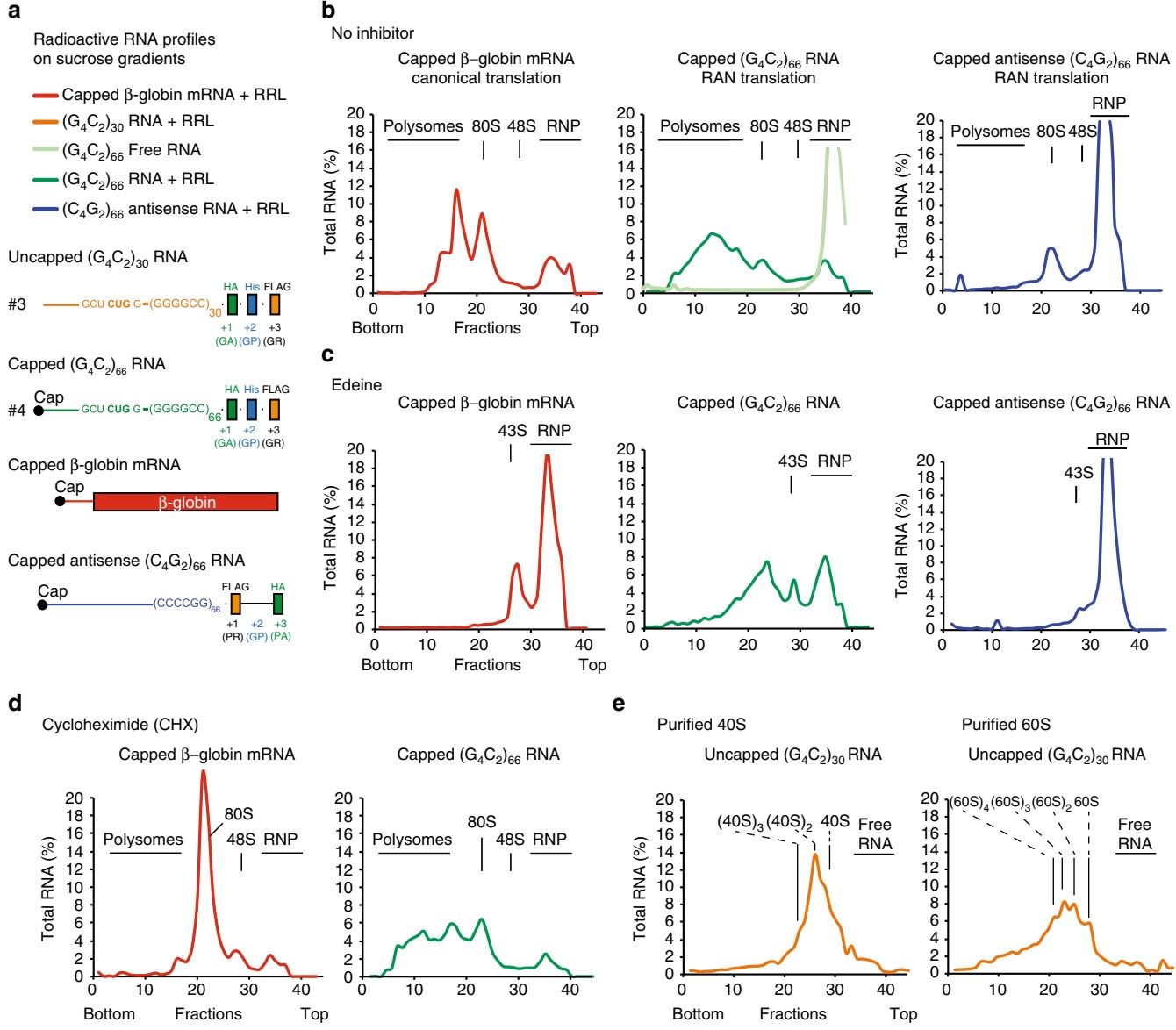

**Fig. 7** $G_4C_2$ containing transcripts have intrinsic ribosome binding capacity independently of their translation. (**a**) Scheme of the capped $(G_4C_2)_{66}$ RNA (#4) and uncapped $(G_4C_2)_{30}$ transcripts (#3) used for translation in RRL and polyribosome fractionation on sucrose gradients. As controls, capped β-globin and capped $(C_2G_4)_{66}$ antisense repeat RNAs were used in the same system. (**b–d**) Radiolabeled capped $(G_4C_2)_{66}$ RNA profile by polyribosome fractionation in RRL comparatively to capped β-globin mRNA and $(C_2G_4)_{66}$ antisense RNAs. Fractionation on sucrose gradients was performed without inhibitor (**b**), in presence of Edeine (**c**) or CHX (**d**). (**e**) Sucrose gradient fractionation of radiolabeled uncapped $(G_4C_2)_{30}$ transcripts (#3) was performed in presence of purified 40S or 60S ribosomal subunits

study RAN translation and determine specific *cis*-requirements and *trans*-requirements for expanded $G_4C_2$ translation. $G_4C_2$ RAN translation was found to share many aspects with canonical translation initiation, including the requirement of a 5′ cap structure, methionylated initiator tRNA[Met], and the recruitment of the 40S subunit by the eIF4F complex (eIF4A, E, and G) to begin scanning toward the start codon (Fig. 8a, b). These findings are consistent with mechanisms involved in RAN translation of CGG triplet repeats in the fragile X *FMR1* gene which also depends on a cap-dependent scanning mechanism[15,33,54]. Since eIF4F's functions were shown to be critical in dysregulation of the translational machinery in cancers, major efforts have been undertaken to develop specific compounds directed against its components for therapeutic purposes[55]. Our work highlights the importance of eIF4F in ALS/FTD pathogenesis, thereby opening

the potential for new therapeutic strategies using existing eIF4F inhibitors to mitigate the effects of this neurodegenerative disease.

Ribosome profiling on higher eukaryotes showed that translation occurs on numerous ORFs without an AUG-initiator but operates with near-cognate start codons (CUG > GUG > UUG > ACG > others)[56,57]. We discovered that the CUG codon located 24 nucleotides upstream of the $G_4C_2$ repeat, in the +1 (GA) frame and in an optimal Kozak sequence, is utilized as a start codon to produce DPR proteins. Mutations of this CUG codon or the Kozak sequence abolish production of all three DPR proteins in RRL supporting a frameshifting model where the ribosome starts at the CUG and slips to translate GP (+2) and GR (+3) (Fig. 8b). As in RRL, RAN translation in all three frames was affected by mutation of the CUG codon in human neural progenitor, mouse motor neuronal cells and HEK293T cells. However, while poly-

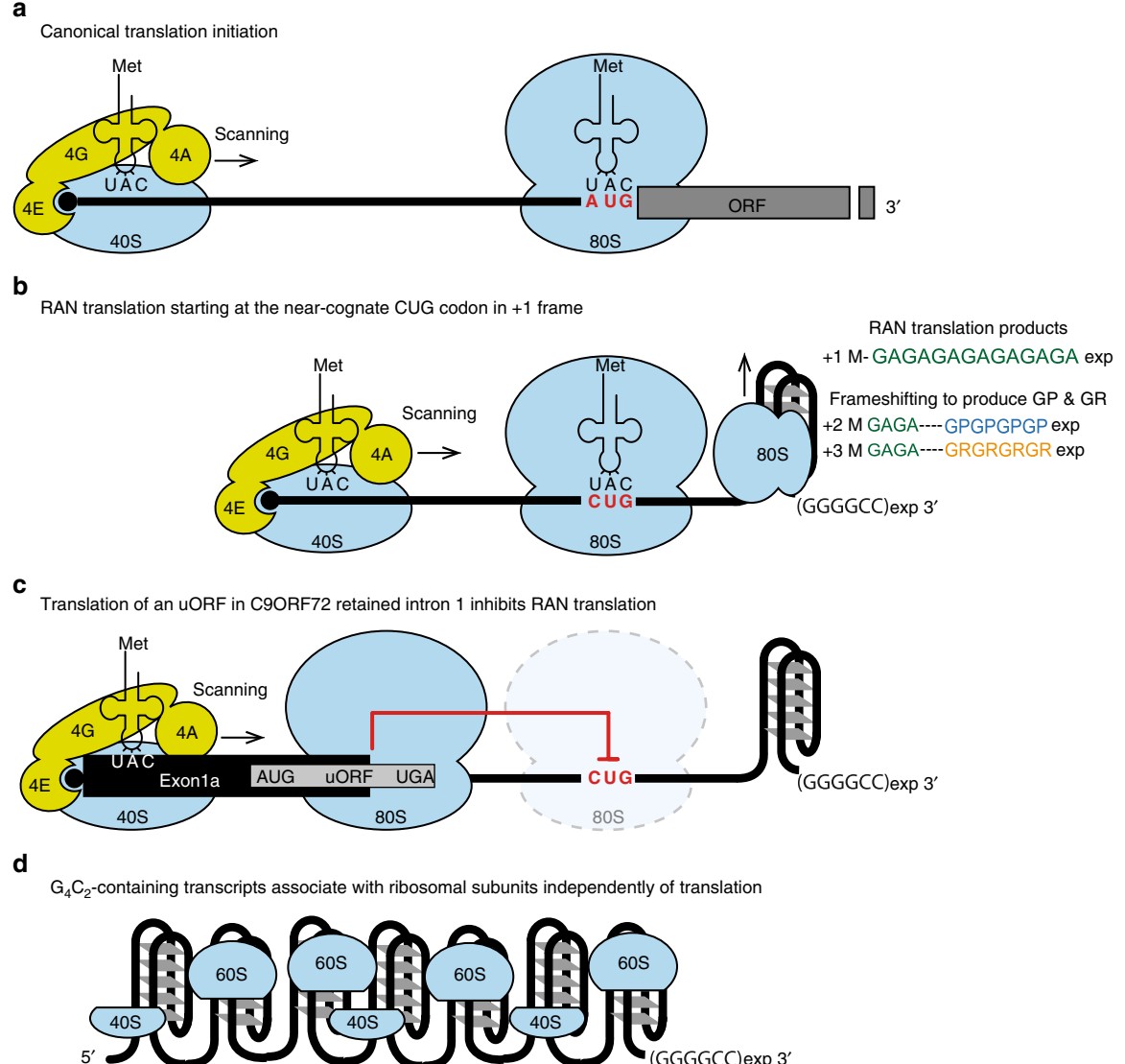

**Fig. 8** Model of translation mechanisms associated with $G_4C_2$ expansions in *C9ORF72* ALS/FTD. (**a**) Pre-Initiation ribosomal complex (PIC) assembles on the 5′ cap of mRNA by interacting with eIF4F complex formed by the cap binding factor eIF4E, the platform eIF4G and the RNA helicase eIF4A. The PIC complex scans the 5′ end for an appropriate AUG start codon, where the 60S ribosomal subunit joins the 40S to form a functional 80S ribosome ready to translate the coding sequence. (**b**) $G_4C_2$ RAN translation initiation shares the same pathway as the canonical one to translate poly-GA dipeptides, including the need of 5′ cap, eIF4E, eIF4G, eIF4A, initiator methionyl-tRNA, and the scanning mechanism. However, it initiates on a near-cognate CUG codon embedded in a perfect Kozak sequence, in frame with poly-GA, instead of a canonical AUG start codon. The ability of $G_4C_2$ expansions to form stable G-quadruplex structures forces the ribosome to occasionally undergo frameshifting to translate poly-GP and poly-GR in the +2 and +3 frames, respectively. (**c**) When $G_4C_2$ repeats are expanded, a subset of *C9ORF72* mRNA is mis-spliced retaining intron 1 with the repeats[35]. RAN translation from these RNAs is inhibited by a uORF that is translated canonically. (**d**) $G_4C_2$ expanded transcripts associate with ribosomal subunits independently from their translation

GP translation was prevented by mutation of the CUG repeat in RRL and the two neuronal models, poly-GP levels were slightly increased in HEK293T cells supporting a context-dependent regulation that differs between the three frames when the CUG is absent. The presence of specific RNA helicases might explain the differences on poly-GP translation between the different cell types, such as DDX21 recently shown to unfold RNA G-quadruplex structures in HEK293T[58]. Notably, an UAG stop codon in phase with the poly-GP frame is present at the beginning of the $G_4C_2$ repeats (UAG GGG CC sequence, Supplementary Fig. 1), indicating that the ribosome must initiate in another reading frame and then frameshift to produce poly-GP or directly initiate within the repeat. As we observed a single band on SDS-PAGE with comparable molecular weight between all reading frames and systems used (Figs. 3 and 4), initiation further

downstream inside the repeats is less likely to occur from $(G_4C_2)_{66}$ transcripts. When comparing translation efficiencies for the three reading frames, poly-GA (+1) is predominant, followed by poly-GP (+2) and poly-GR (+3), which is in agreement with a frameshifting model. This is also consistent with staining and immunoassay from human post-mortem tissues, where poly-GA accumulates at higher levels than poly-GP and poly-GR (Supplementary Fig. 3)[17,40,59,60].

$G_4C_2$ RAN translation initiation is influenced by repeat length, with different sensitivity among the three reading frames. While RAN translation efficiency is only reduced in the +1 poly-GA and +3 poly-GR frames with shorter repeat length, it is completely abolished for poly-GP at 30 comparatively to 66 repeats (Fig. 1). This repeat length dependence could reflect secondary structures, which differentially affect ribosomal scanning, translation

elongation or force the ribosome to undergo a frameshifting. Indeed, $G_4C_2$ expansions can adopt RNA G-quadruplexes[28–32], a structure that was recently demonstrated to induce frameshifting during translation[61,62]. These RNA secondary structures are stable in presence of monovalent cations, in the order of K+ > Na+ > Li+ [63]. Thus, variations of ion concentration in the cell or specific RNA binding proteins[58] may modulate the presence of G-quadruplex structure in $G_4C_2$-containing transcripts and could influence frameshifting or initiation at non-AUG start codon.

Another major finding is the down-regulation of $G_4C_2$ RAN translation by a short uORF. Indeed, in mis-spliced C9ORF72 transcripts that retain intron 1, an uORF is present with an AUG and two in-frame stop codons located 76 nucleotides upstream of the $G_4C_2$ repeats. Notably, the AUG codon in exon 1A is the only AUG identified in the 5′end of the mis-spliced RNA. Upstream ORF are cis-acting elements that regulate the expression of downstream protein coding sequences[25,45,46]. We demonstrated that mutating the AUG start codon of the uORF is sufficient to increase $G_4C_2$ RAN translation in all three reading frames, confirming that this uORF is efficiently used by the ribosome during 5′–3′ scanning and is therefore inhibiting the translation of the downstream $G_4C_2$ repeat (Figs. 5 and 8c). It is noteworthy that translation of synaptic mRNA(s) was shown to be downregulated by uORF(s) located in their 5′UTR, but upregulated upon metabotropic glutamate receptor activation[64–66]. Thus, it will be important to determine whether the uORF in mis-spliced C9ORF72 transcripts influences $G_4C_2$ RAN translation level upon synaptic activation or external stimuli in neurons.

Notably, ASOs directing RNase-H-dependent degradation of C9ORF72 transcripts are under therapeutic development[5–7,52]. The identification of sequences upstream of the repeat that influence RAN translation (CUG near-cognate start codon and uORF) opens the possibility of using alternative strategies based on ASOs that modulate translation without reducing transcript levels[67,68]. In agreement, we demonstrated that several RNA ASOs specifically targeting the region immediately upstream of the repeats block ribosomal scanning and efficiently reduce the level of RAN translation products (Fig. 6f–h).

Finally, we show that $G_4C_2$ repeat transcripts unexpectedly associate with ribosomal subunits in a translation independent manner (Fig. 8d). Indeed, blocking cap initiation factors, codon–anticodon interaction, 80S ribosome assembly and ribosomal elongation did not avert the sedimentation of radiolabeled $G_4C_2$ RNAs in the heavy fractions of sucrose gradients (Fig. 7). In addition, removing the 5′cap, shortening the repeat size, or using purified ribosomal subunits did not prevent the assembly of the transcript to multiple ribosomal subunits. On the contrary, antisense transcripts with $C_4G_2$ repeat did not associate with the ribosome. This striking finding supports a RNA gain-of-function mechanism, independent from RAN translation and DPR proteins accumulation. Ribosomal subunits are assembled in the nucleolus and exported to the cytoplasm by multiple export receptors[69]. It will be important to determine whether sequestration of ribosomal subunits by expanded repeats and disruption of nucleocytoplasmic transport recently identified in C9ORF72 disease[4] negatively impact overall translation in cells with C9ORF72 expansions.

Overall, we provide new insights into RAN translation of C9ORF72 $G_4C_2$ repeat which uses a cap-dependent mechanism initiating at a near-cognate CUG codon. A novel mechanism of toxicity associated to C9ORF72 expansion is supported by the association of $G_4C_2$ transcripts with ribosomal subunits independently of their translation. Importantly, this work identifies sequences upstream of the $G_4C_2$ repeats and specific initiation factors as possible therapeutic targets to inhibit RAN translation in C9ORF72 ALS/FTD patients.

## Methods

**Generation of C9ORF72 constructs with $G_4C_2$ repeats**. To generate the different constructs listed in Supplementary Fig. 1 and Table 1, a plasmid pAG3 containing 66 repeats[20,36] was first digested with restriction sites BssHII and SacI to isolate the intronic region of human C9ORF72 with $(G_4C_2)_{66}$, including 8 bp of 5′, 99 bp of 3′ flanking sequences and three tags in frame with DPR proteins. BssHII is a restriction site naturally present in the human C9ORF72 gene located two nucleotides upstream of the repeats. Second, pUC18 (ThermoFisher, # SD0051) was modified to contain the three HindIII, BssHII, and SacI restriction sites, enabling the insertion of the digested BssHII/SacI C9ORF72 insert and the addition of any 5′end sequence between the HindIII and BssHII sites. After cloning the C9ORF72 insert in modified pUC18 with BssHII and SacI, primers listed in Supplementary Table 2 were used to generate the different constructs listed in Supplementary Fig. 1. Primers were designed to add the T7 Promoter for in vitro transcription (construct #9), followed immediately by 113 bp of 5′ flanking $G_4C_2$ sequence with CUG > CCG mutation (construct #5), GAG > GGG mutation (construct #6) and double mutations CUG > CCG + GAG > GGG (construct #7). Also, primers were designed to add T7 promoter followed by 320 bp of 5′ sequence (construct #1), 320 bp with AUG > CGG mutations (construct #2) and to generate a short 5′end by adding T7 promoter with 33 bp (construct #8). All primers were designed to harbor the HindIII restriction site at the 5′ end and BssHII site at the 3′ end. After phosphorylation with T4 Polynucleotide Kinase (ThermoFisher, #EK0031) of the primers at the 5′end and hybridization of corresponding forward and reverse primers, the generated inserts were cloned in HindIII-BssHII pUC18 with $(G_4C_2)_{66}$ repeats. The original plasmid was modified to contain T7 promoter by cloning using the HindIII restriction site (construct #4). Construct #3 with 30 $G_4C_2$ repeats was generated by expansion retraction during amplification of the construct #4 with 66 repeats. Finally, construct #5 containing CUG > CCG mutation was digested with HindIII and NotI to be cloned in pAG3 downstream of the CMV early enhancer/chicken β-actin (CAG) promoter for human cell transfection.

The $C_4G_2$ antisense construct used as control in Fig. 7 was cloned by digesting pAG3 containing 66 repeats[20,36] with restriction sites BssHII and NotI to isolate the intronic region of human C9ORF72 with $(G_4C_2)_{66}$ and cloning it into puc18 harboring T7 promoter in antisense direction. This construct was designed to harbor Flag tag in poly-PR +1 frame and HA tag in +3 poly-PA frame.

**In vitro transcription**. The different variants of C9ORF72 $(G_4C_2)_{exp}$ constructs were cloned downstream of T7 promoter in pUC18 as detailed in Supplementary Fig. 1 and Table 1. Vectors were digested by XhoI for run-off in vitro transcription with T7 RNA polymerase. Uncapped RNAs were separated on denaturing PAGE (4%) and RNA were recovered from the gel slices by electro-elution. The resulting pure RNA transcripts were capped at their 5′ end with the ScriptCap $m^7G$ Capping System (Epicenter Biotechnologies).

**In vitro translation in RRL**. Translation reactions were performed in self-made rabbit reticulocyte lysate system (RRL) as previously described[42], without RNase treatment (used in commercially available extracts) that was shown to be detrimental to the translation efficiency from extracts, especially for cap-dependent translation[70]. Briefly, reactions were incubated at 30 °C for 60 min and included 100 and 200 nM of each transcript and 10.8 μCi [$^{35}$S]Met. Aliquots of translation reactions were analyzed by 15% SDS-PAGE and Western Blots. The cap dependency was analyzed by preincubation of increasing $m^7$GpppG concentrations ranging from 0.5 to 1.5 mM for 5 min at room temperature. The experiments were performed in the presence of $MgCl_2$ at a constant $[MgCl_2]/[cap\ analog]$ ratio of 0.8. For translation in presence of RNA sense (RNA-SO) and antisense (RNA-ASO) oligonucleotides (Supplementary Table 3) were annealed to 100 nM capped 66 repeat RNA (construct #4) in 20 mM Hepes-K (pH 7.6) and 100 mM KC1 for 5 min at 65 °C and 20 min at room temperature with a 10 or 50 fold molar excess of oligonucleotides over construct #4. This annealing mixture was kept on ice before addition to the translation reaction. RRL were incubated 5 min at 30 °C in presence of the different translational inhibitors at the following concentrations: 150 ng mL$^{-1}$ for the polyI:C, 15 μM for salubrinal, 4.5 mg mL$^{-1}$ cycloheximide, 10 μM edeine, 15 μM FL3, and 5 μM 4E1RCat.

**Sucrose-gradient analysis**. For sucrose-gradient analysis, 5′-$^{32}$P-labeled or 3′-$^{32}$P-labeled mRNA were incubated in RRL or with purified 40S and 60S ribosomal subunits, in the presence of specific inhibitors (Edeine leads to 43S accumulation, GMP-PNP leads to 48S formation, cycloheximide blocks translocation and leads to 80S accumulation) or without inhibitor to assemble polysomes. Translational inhibitors were incubated with RRL 5 min prior to addition of radiolabeled mRNAs. The translation initiation complexes were separated on a 7–47% linear sucrose gradient in buffer (25 mM Tris–HCl [pH 7.4], 50 mM KCl, 5 mM $MgCl_2$, 1 mM DTT). The reactions were loaded on the gradients and spun (23,411×g for 2.5 h at 4 °C) in a SW41 rotor. mRNA sedimentation on sucrose gradients was monitored by Cerenkov counting after fractionation. In Supplementary Fig. 6i, capped $(G_4C_2)_{66}$ transcripts were folded in presence of KCl, NaCl or LiCl at 195 mM, by denaturing 1 min at 95 °C, followed by 5 min at 20 and 4 °C until adding the RRL (75 mM final ion concentrations).

**Cell culture and plasmid transfection**. The HEK293T cells were cultured in DMEM 10% (v/v) FBS and penicillin/streptomycin. ReNcell VM human neural progenitors (Millipore; Catalog number SCC008) were maintained in high-glucose DMEM/F12 (ThermoFisher Scientific) media supplemented with 2 µg mL$^{-1}$ heparin (StemCell Technologies, #07980), 2% (v/v) B27 neural supplement (ThermoFisher Scientific, #175004044), 20 µg mL$^{-1}$ hEGF (Sigma-Aldrich, #E9644), 20 µg mL$^{-1}$ bFGF (Stemgent, #03-0002) and 1% penicillin/streptomycin (ThermoFisher Scientific) and were plated onto BD Matrigel (BD Biosciences)-coated cell culture flasks with B27, EGF, FGF, and heparin on precoated Matrigel dishes. The NSC-34 cells (CELLutions Biosystems Inc; Catalog number—CLU140) were grown in DMEM supplemented with 10% FBS, 100 U mL$^{-1}$ penicillin, and 100 µg mL$^{-1}$ streptomycin at 37 °C in a humidified atmosphere of 5% CO$_2$. HEK293T were plated 24 h prior transfection with different *C9ORF72* (G$_4$C$_2$)$_{66}$ expansion constructs (Supplementary Fig.1 and Table 1) and a reporter pGFPmax (Lonza) expressing GFP using a construct:pGFPmax ratio of 5:1. The lipofectamine 2000 reagent was used according to manufacturer instruction (Invitrogen) for HEK293T and NSC-34 transfections. Nucleofection using Nucleofector kit (Lonza, #VPG 1005) was used for neural progenitor cell to achieve high efficiency of transfection of plasmids. Twenty-four hours after transfection, the cells were washed with PBS 1X and collected for RNA and protein extractions.

**FL3 treatment in cells**. HEK293T were cultured 24 h prior treatment into 10 cm dish, following by transfection with lipofectamine 2000 of construct #4 as described previously. After 4 h of incubation in the transfection reagents, cells were treated with 10 µM FL3 for 24 h and collected for immunoblot analysis

**Immunoblotting**. The cell pellets were re-suspended in 400 µl of 2X Laemmli sample buffer (Biorad #1610737). The proteins were homogenized with pestle, then denatured at 95 °C for 10 min. The total protein extract was separated on gradient 4–20% SDS-PAGE gels and 18% SDS-PAGE gels, transferred onto PVDF membranes, blocked with 5% (v/v) non fat dry milk (NFM) in Tris–buffered saline (TBS) pH 7.5. The membranes were incubated with primary antibodies (Supplementary Table 4) overnight at 4 °C in TBS and 5% (v/v) NFM, washed with TBS-Tween 20 0.1%, incubated with horseradish peroxidase (HRP)-conjugated secondary antibodies (donkey anti-rabbit GE Healthcare Life Sciences #NA934, sheep anti-mouse GE Healthcare Life Sciences #NA931, goat anti-rat abcam #97057), washed with TBS-Tween 20 0.1% and signal was revealed with autoradiographic films.

**Immunofluorescence**. HEK293T cells were cultured on 24-well plates prior transfection with lipofectamine 2000, following the recommendations of supplier. Twenty-four hours after transfection, the cells were fixed in 4% paraformaldehyde and washed twice with PBS. Cells were permeabilized with 0.1% Triton X-100 for 10 min at room temperature. They were washed twice again with PBS and blocked with 1% bovine albumin in PBS for 1 h at room temperature. Cells were incubated at 4 °C for 24 h with primary antibodies anti-GA or anti-GP (Supplementary Table 4) at 1:500 dilution in the blocking solution supplemented with 0.02% Tween-20. Rabbit fluorescently tagged secondary antibody conjugated to Alexa 595 (ThermoFisher Scientific) was incubated for 1 h at room temperature in the blocking buffer. The nuclei were stained with ProLong™ Gold Antifade Mountant with DAPI (ThermoFisher, # P36935) and mounted on slides for confocal microscopy.

**Immunohistochemistry of human brain sections**. Paraffin sections (8 µm) from the cerebellum were deparaffinized with CitriSolv (Thermo Fisher Scientific, #04-355-121) and incubated in 100% EtOH, 90% EtOH, 70% EtOH, 50%, and Milli-Q® water. Sections were incubated in 0.6% H$_2$O$_2$ in methanol at room temperature for 15 min, treated with antigen unmasking solution (Vector Laboratories, #H-3300) in the steam chamber for 45 min, and blocked at room temperature with 1% FBS/0.1% Triton X-100/PBS for 25 min. Sections were then incubated at 4 °C overnight with anti-GA rabbit antibody (Rb4334) (1:1000), anti-GR rabbit antibody (Rb4995) (1:1000), or anti-GP rabbit antibody (Rb7633) (1:1000)[52] diluted in 1% FBS/PBS. Next, sections were stained with secondary antibody ImmPRESS™ HRP (peroxidase) anti-Rabbit IgG Reagent (Vector Laboratories, #MP-7401) at room temperature for 1 h, developed with VECTOR NovaRED Peroxidase (HRP) Substrate Kit (Vector Laboratories, #SK-4800), treated with hematoxylin stain solution (RICCA, #3530-32) and bluing reagent Scott's tap water substitute (Leica Biosystems, #3802901), and mounted with Richard-Allan Scientific™ Mounting Medium (Thermo Fisher Scientific, #4112).

**Data availability**. The data that support the findings of this study are available from the corresponding author upon request. All constructs and reagents generated in this study will be shared upon request.

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

## Acknowledgements

We thank Dr Doo Yeon Kim for his support in culturing ReNcell VM, Amélie Laugel, Michael Baughn, Dr Anna-Claire Devlin, and Dr Ying Sun for technical assistance, Dr Gilbert Eriani, members of Dr Brian J. Wainger and Dr Mark W. Albers laboratories, Dr Merit Cudkowicz, Dr Shuying Sun, Dr Raymond Kaempfer, Dr Frank Rigo, and Dr Don W. Cleveland for helpful discussions and continuous support. R.T. was supported by a grant from the Philippe Foundation. This work was supported by CNRS, Université de Strasbourg, a grant from the ANR to F.M. (ANR-11-SVSE802501). C.L.-T. was supported by the Department of Neurology at the Massachusetts General Hospital and grants from Target ALS (13-04827) and from NINDS/NIH (R01NS087227).

## Author contributions

R.T., F.M. and C.L.-T. designed research; R.T., F.F., F.M. and C.L.-T. analyzed the data; R. T., LS., F.M., F.F., M.J., M.W., C.-Z.L., C.-C.L. and T.G. performed research; J.J, L.D., H. A.-H., K.J.-W., T.G. and L.P. contributed key reagents and methodology. R.T., F.F., F.M. and C.L.-T. wrote the manuscript.
