## [Peer Review File · Nature Communications]

Reviewers' comments:

Reviewer #1 (Remarks to the Author):

Hexanucleotide expansions (G4C2)_{exp} in the C9ORF72 gene are the major cause of two fatal neurodegenerative disorders, ALS and FTD. In this study, Tabet et al. recapitulated and investigated repeat-associated non-AUG (RAN) translation of the human C9ORF72 expansion transcripts in a rabbit reticulocyte lysate (RRL). This translation occurs from all reading frames and requires a CUG start codon and an initiator Met-tRNA_i. Mutations of the CUG start codon affected synthesis of proteins from all the three reading frames. An upstream open reading frame in mis-spliced C9ORF72 transcript is shown to be inhibitory for RAN translation. Surprisingly, and in contrast to conventional mRNAs, the expanded transcripts bind ribosomal subunits independent from their translation.

Overall, this paper presents a detailed and comprehensive analysis of RAN translation directed by C9ORF repeat expansion, which is a frequent cause of ALS/FTD neurodegenerative disorders. However, there are several questions, some of them should be experimentally addressed.

Comments

1. (General). It is very puzzling that the products of RAN translation could be labelled with [35S] Methionine in an RRL (Figs. 2b, e and 3c). The lysate contains methionine aminopeptidase that co-translationally cleaves the initiator methionine when the nascent polypeptides are ~20 residues long. Usually, the initiator N-formyl[35S] Met-tRNA_i, rather than [35S] Methionine is used in this type of experiments. This tRNA labels only the NH₂-terminus of proteins, and the formyl group prevents the action of the aminopeptidase. Unless incorporation from N-formyl[35S] Met-tRNA_i is shown, they could not conclude unequivocally about the role of Met-tRNA_i in translation initiation.
2. Figure 1a-c. What is the ~14 kDa translation product? (Legend) It is not clear what concentrations of mRNAs were used.
3. Figure 2B. The mRNA minus control is missing. In addition, their system exhibits very high translation of endogenous globin mRNA. Wasn't the lysate treated with micrococcal-nuclease?
4. Figure 3. It would be interesting to investigate the role of the optimal -3/+4 nucleotide context in RAN translation. In addition, one wonders whether the substitution of the canonical initiator AUG for CUG in construct 4 can enhance translation. The authors might have these data already.
5. Figure 4b. It is bothersome that mutating the CUG codon to CCG does not inhibit the production of poly-GP in HEK293 cells, as it does in RRL (Figure 3d). The authors suggest that in cells the poly-GP translation occurs from an alternative start codon and is influenced by trans-acting factors that are absent in RRL. This raises questions about the relevance of the use of the RRL system to study the translational mechanism. Could they rescue poly-GP synthesis from the mutated construct in RRL by adding an extract from HEK293 cells?
6. Figure 7. It is concluded that the G4C2 containing transcripts sequester ribosomal subunits, and presumably inhibit global translation. This result is rather preliminary. Could they test this prediction by exploring the effect of these transcripts' sequestration, in trans on the translation of a reporter mRNA in RRL?
7. Figure 7e shows the position of RNP. This makes no sense as their assays were done only with mRNA and ribosomal subunits. Do I miss something?

Minor comments:

1. Page 3 "...process that requires numerous elongation initiation factors (eIFs)." should read "...process that requires numerous eukaryotic initiation factors (eIFs)."

Reviewer #2 (Remarks to the Author):

This manuscript examines the ALS/FTD C90rf72 gene and its expression as RAN translation. From the studies conducted, the authors conclude that RAN translation of this mRNA occurs in a cap-dependent manner utilizing a CUG codon for initiation. They also find that these transcripts are sticky/bind to either 40S or 60S subunits.

Major concerns

1. The authors use the IGR from the Cricket paralysis virus as a control for "efficient translation". However, it would seem more appropriate to use a normal, cap-dependent reporter to see just how efficiently the repeat transcripts are expressed (i.e. a globin mRNA derivative). Secondly, it is curious that there appears to be little dependency on mRNA input with only the GR product showing an increase with increased RNA (66) although there is a decrease with added RNA for the 30 repeats. Third, as relates to the relative synthesis of either GA, GP or GR, is it possible that this reflects the tRNA populations present in RRL? An examination of the rabbit beta globin chain mRNA indicates the following use of codons that might arise from the G4C2 repeat: arginine – no CGG codons used; alanine – half of the codons used are GCC; proline – no CCG codons used; glycine – more than half of the codons used are either GGG or GGC. If one assumes that in the reticulocytes, which are synthesizing 95-98% hemoglobin, that the tRNA population is a match to the amino acids in hemoglobin, might this then be reflected in the synthesis seen in Figure 1, panels A, B and C?
2. A more convincing proof that RAN translation initiates with methionine would be to add poly(IC) to activate PKR and show that phosphorylation of eIF2 reduces expression of the peptides. It is noted even for globin synthesis that the N-terminal, initiating methionine is removed and thus the only methionine registered is from an internal methionine. Based upon the N-end rule, the amino acid coded for following the CUG codon would be glutamic acid (GAA) and this should result in the removal of the N-terminal methionine (see Huang et al. Biochemistry 1987).
3. The authors do show that the translation of their transcripts is favored when the mRNA is capped. However, what is the evidence that in vivo the mRNA responsible for RAN translation is capped?
4. What is the evidence that the CUG initiating codon does in fact direct the binding and use of Met-tRNA_i (see above concern)? The use of CUG codons and leucyl-tRNA have shown up in several recent publications.
5. Figure 7 – binding of GGGGCC transcripts to ribosomal subunits. This experiment is uncontrolled. It would appear that the transcripts are being bound non-specifically. Controls such as globin mRNA or the IRG segment used in Figure 1 should be used to ensure that the observed binding is of some relevance. This is especially worrisome for the apparent "polysome-like" aggregates seen in Panel E.

Minor concerns

1. The authors would benefit from reviewing their manuscript for better use of English and to remove some technical errors (i.e. Introduction – "... that requires numerous elongation initiation factors (eIFs)..." The e in eIF stands for eukaryotic, not elongation.).
2. Why is the level of cap inhibitor (m7GpppG) so high to affect inhibition (1.5 mM). Often in other studies, the level used was in the 100 micromolar range. This level represents a 15,000 to 1 ratio of analog to mRNA.

3. Figure 2 – it is not clear from the transcripts whether there are any internal methionines in the coding region. This should be checked and reported.

Revision of the manuscript “ALS/FTD *C9ORF72* transcripts initiate translation at a CUG codon and sequester ribosomal subunits” (NCOMMS-17-10398) submitted by Tabet et al.

General response to the Editor and Reviewers

We are grateful for the overall positive feedback from the referees and addressed all editorial and reviewers’ concerns to improve the quality of our study. In summary, we have provided the following new pieces of evidence to strengthen our initial findings:

- The main concerns raised were related to the methionine incorporation in DPR products and were in part due to imprecisions in our initial description of the constructs. We have now clarified in the text that the G_4C_2 repeat constructs used to monitor RAN translation and incorporation of ^{35}S -methionine do not contain any AUG codon in none of the three frames and the DPR products should not incorporate any methionine other than at the initiation codon. Consistently, we observed incorporation of radiolabelled methionine in DPR products translated from constructs harboring a near cognate CUG start codon located in the +1 frame (poly-GA). The translation of ^{35}S -methionine DPR products was alleviated by a single mutation of the CUG codon into CCG, indicating that the methionine is incorporated at the N-terminal of the DPR proteins. We reinforced our finding by replacing the CUG into a canonical AUG codon and by mutating the surrounding Kozak sequence, as suggested by Reviewer 1 (**new Figure 3e,f**). As expected, a start codon AUG instead of CUG increases the level of methionine incorporation and the levels of all three DPRs. In contrast, mutation of the Kozak sequence inhibits the production of ^{35}S -methionine proteins, poly-GA, poly-GP and poly-GR, supporting a frameshifting mechanism where translation of all DPR proteins starts at the CUG codon in the +1 frame and undergoes frameshifting to produce poly-GP and poly-GR. Furthermore, inhibiting the ternary complex eIF2-methionylated-tRNA-initiator by Poly(I:C), as suggested by Reviewer 2, blocks ^{35}S -methionine incorporation demonstrating the role of methionylated-tRNA-initiator in DPR translation initiation (**new Supplementary Figure 4a,b**).
- We strengthened our findings by demonstrating that the CUG near cognate start codon plays a crucial role in *C9ORF72* RAN translation *in vitro* (RRL) as well as in human cells including HEK293 and human neural progenitors (**new Figure 4**). Indeed, mutating the CUG codon into CCG altered RAN translation in all three systems (RRL, HEK293 and human neural progenitors), confirming that RAN translation starts at CUG codon and undergoes frameshifting *in vivo*. Interestingly, translation in the poly-GP frame was differently regulated in HEK293 cells compared to RRL and neural progenitors suggesting cell type-specific mechanisms of translation regulation.
- We also confirmed that cis-acting elements in the 5’ flanking sequence are important for RAN translation control in *C9ORF72* patient fibroblasts. Indeed, in our initial manuscript we had demonstrated that the translation of an upstream open reading frame (uORF) inhibits RAN translation of the downstream G_4C_2 repeat. Following an approach recently developed for other uORFs (Liang et al., *Nature Biotechnology*, 2016), we now show that blocking the AUG region of the upstream ORF with chemically modified antisense oligonucleotides that do not trigger RNase H cleavage increases the level of RAN translation in patient cells (**new Figure 5e,f**).

- We complemented our study with several additional experiments suggested by the reviewers such as comparing RAN translation efficiency in our RRL system not only to IRES-dependent translation but also to the canonical cap scanning mechanism (**new Supplementary Figure 2**). We also compared G₄C₂ repeats RNA profile in polyribosomes purification to the antisense C₄G₂ repeats RNA, and we observed that only G₄C₂ RNA is capable to sequester ribosomal subunits independently of RAN translation (**new Figure 7a-c**).

We provide below a point by point response to the Reviewers and hope that the revised manuscript can now be recommended for publication in *Nature Communications*.

Reviewer #1:

Hexanucleotide expansions (G4C2)_{exp} in the C9ORF72 gene are the major cause of two fatal neurodegenerative disorders, ALS and FTD. In this study, Tabet et al. recapitulated and investigated repeat-associated non-AUG (RAN) translation of the human C9ORF72 expansion transcripts in a rabbit reticulocyte lysate (RRL). This translation occurs from all reading frames and requires a CUG start codon and an initiator Met-tRNA_i. Mutations of the CUG start codon affected synthesis of proteins from all the three reading frames. An upstream open reading frame in mis-spliced C9ORF72 transcript is shown to be inhibitory for RAN translation. Surprisingly, and in contrast to conventional mRNAs, the expanded transcripts bind ribosomal subunits independent from their translation.

Overall, this paper presents a detailed and comprehensive analysis of RAN translation directed by C9ORF repeat expansion, which is a frequent cause of ALS/FTD neurodegenerative disorders. However, there are several questions, some of them should be experimentally addressed.

Comments

1. (General). It is very puzzling that the products of RAN translation could be labelled with [35S] Methionine in an RRL (Figs. 2b, e and 3c). The lysate contains methionine aminopeptidase that co-translationally cleaves the initiator methionine when the nascent polypeptides are ~20 residues long. Usually, the initiator N-formyl[35S] Met-tRNA_i, rather than [35S] Methionine is used in this type of experiments. This tRNA labels only the NH₂-terminus of proteins, and the formyl group prevents the action of the aminopeptidase. Unless incorporation from N-formyl[35S] Met-tRNA_i is shown, they could not conclude unequivocally about the role of Met-tRNA_i in translation initiation.

We agree with the reviewer that the N-terminal methionine is usually processed. However, using the histone H4 mRNA (which contains the initiator methionine and a single internal methionine), we have previously determined that in our self-made Rabbit Reticulocyte lysates, only ~60-70% of the N-terminal methionine is processed leaving a residual N-terminal ³⁵S-methionine (Martin et al., 2011; Martin et al., 2016). An incomplete processing of the N-terminal methionine was also found *in vivo* with ~20% of the N-terminal methionine being acetylated instead of removed (Gigliione et al., 2015). In this manuscript, none of the constructs, #3 to #11, used to study the G₄C₂ translation contain any AUG codon in none of the three frames (**Supplementary Fig. 1; Table S1**). Hence, the DPR products do not incorporate any internal methionine and the radiolabelled DPR products that we observe at the expected size (**Figure 2**) derive from the incorporation of N-terminal ³⁵S-methionine. This

has now been clarified in the text and we have included a control without G₄C₂ repeats to confirm that the ³⁵S-labelled product is dependent on the translation of the G₄C₂ repeat transcripts (**new Figure 2b**). Consistently, ³⁵S-labelled peptides are immunoprecipitated by an antibody against the HA tag which is in frame with the poly-GA dipeptide repeat proteins (**Figure 2c**).

In addition, it is increasingly recognized that processing of the N-terminus is influenced by the nature of the second amino-acid (Frottin et al., 2006; Martinez et al., 2008). We used the “Terminator” software available on line (<https://bioweb.i2bc.paris-saclay.fr/terminator3/>) to predict the N-terminus of mature poly-GA and control renilla luciferase proteins. The poly-GA N-terminus is predicted with a likelihood of 100% to be a methionine, meaning unprocessed, due to the presence of a glutamic acid residue in the second position, while the methionine of the renilla luciferase N-terminus is predicted to be processed with a likelihood of 77%.

Proteins	N-terminal sequence or entry code (first 20 characters)	Predicted N-terminus of the mature protein	Likelihood (%)	Translation efficiency	Predicted Half-life (hours)
Renilla Luciferase	MTSKVYDPEQRKRMITGPQW	Ac-T(2)	77	1	65
poly-GA	MELRSRALGAGAGAGAGAGA	Ac-M(1)	100	5	5-31

Table 1. Processed N-termini prediction of mature poly-GA products by the software Terminator (Frottin et al., Mol. Cell 2006; Martinez et al., Proteomics 2008)
<https://bioweb.i2bc.paris-saclay.fr/terminator3/>
 The Methionine is numbered M(1) and the following amino-acids are numbered from 2 to 20 with a total size of 20 amino-acids
 The most likely amino-acid to be at the N-terminus is in “Predicted N-terminus of the mature protein”, with the corresponding “Likelihood” percentage.

Finally, following the suggestions from both reviewers we have performed 2 new experiments further confirming that RAN translation of G₄C₂ initiates by a methionine.

First, we generated two additional mutants of our G₄C₂ repeat transcripts (see also comment #3 of Reviewer 1). In the first mutant construct, replacement of the CUG by a genuine AUG start codon leads to the production of DPR proteins with a similar size compared to the CUG native transcript indicating that radiolabeled DPR proteins result from initiation at this position with incorporation of ³⁵S-methionine. In the second one, the Kozak sequence of the CUG has been mutated which dramatically reduced the labeling of the DPR providing another evidence that initiation takes place at this codon with a ³⁵S-methionine residue (**new Figure 3e,f**).

Second, we inhibited the ternary complex formation by inducing the phosphorylation of eIF2 alpha subunit (poly (I:C)/salubrinal treatment) thereby inhibiting the recruitment of methionylated initiator tRNA (see also comment #2 of Reviewer 2). Such a treatment inhibits ³⁵S-methionine incorporation of canonical cap-dependent translation but not IGR-driven translation which does not use the initiator tRNA to start translation. This treatment also inhibits ³⁵S-Methionine incorporation in DPR products, confirming the methionine incorporation by methionylated initiator tRNA and the involvement of eIF2 in RAN translation (**new Supplementary Fig. 4**).

Altogether, these results demonstrate that synthesis of the detected peptides is indeed starting by a methionine residue.

References related to this comment:

- Martin F, Barends S, Jaeger S, Schaeffer L, Prongidi-Fix, L and Eriani G. Cap-assisted internal initiation of translation of histone H4. (2011) Mol Cell 41, 197-209.
- Martin F, Ménétret JF, Simonetti A, Myasnikov AG, Vicens Q, Prongidi-Fix L, Natchiar SK, Klaholz BP, Eriani G. Ribosomal 18S rRNA base pairs with mRNA during eukaryotic translation initiation. (2016) Nat. Commun. 7:12622 doi: 10.1038/ncomms12622.

- Giglione C, Fieulaine S, Meinnel T. N-terminal protein modifications: Bringing back into play the ribosome (2015) *Biochimie* 114, 134-146.
- Frottin F, Martinez A, Peynot P, Mitra S, Holz RC, Giglione C, Meinnel T. The proteomics of N-terminal methionine cleavage. (2006) *Mol Cell Proteomics* 5,2336-49.
- Martinez A, Traverso JA, Valot B, Ferro M, Espagne C, Ephritikhine G, Zivy M, Giglione C, Meinnel T. Extent of N-terminal modifications in cytosolic proteins from eukaryotes. (2008) *Proteomics* 14, 2809-31.

2. Figure 1a-c. What is the ~14 kDa translation product? (Legend) It is not clear what concentrations of mRNAs were used.

The ~14 kDa translation product corresponds to a polypeptide expressed from a transcript containing 30 G₄C₂ repeats. We confirmed that the band corresponds to the expected product by immunoprecipitation with an antibody specific for HA-tag (in frame with poly-GA). The concentrations of RNA used in the translation experiments are ranging from 100 to 200 nM. This information has now been included in the legend.

3. Figure 2B. The mRNA minus control is missing. In addition, their system exhibits very high translation of endogenous globin mRNA. Wasn't the lysate treated with micrococcal-nuclease?

We thank the reviewer for pointing out this omission. We have now included ³⁵S-Met autoradiograph corresponding to a translation experiment in RRL with and without G₄C₂ RNA (negative control) (**new Figure 2b**). As expected, the specific ³⁵S-labelled band immunoprecipitated by anti-HA antibody (Fig. 2c) is not observed in absence of G₄C₂ RNA (66 repeats, construct #4).

Our self-made rabbit reticulocyte lysates are indeed not treated by micrococcal-nuclease. Therefore, we can observe on ³⁵S-Met autoradiographs the synthesis of endogenous globin and lipoxygenase that are the most prevalent mRNAs in reticulocyte lysates. This method enables us to monitor in our extracts the translation efficiency of an internal endogenous control. In addition, the lack of nuclease treatment provides more physiologically relevant cell-free translation extracts that recapitulate more faithfully *in vivo* translation features such as cap-dependency as previously described (Ricci et al., 2011).

Reference related to this comment:

Ricci EP, Limousin T, Soto-Rifo R, Allison R, Pöyry T, Decimo D, Jackson RJ, Ohlmann T. Activation of a microRNA response in trans reveals a new role for poly(A) in translational repression. (2011) *Nucleic Acids Res.* 39, 5215-31.

4. Figure 3. it would be interesting to investigate the role of the optimal -3/+4 nucleotide context in RAN translation. In addition, one wonders whether the substitution of the canonical initiator AUG for CUG in construct 4 can enhance translation. The authors might have these data already.

We are grateful for this suggestion and, as described in the first comment, we have performed these experiments. Mutating the CUG near cognate start codon into a genuine AUG significantly increased RAN translation in all reading frames (**new Figure 3b,e,f** and **Supplementary Fig. 1 construct #10**), confirming that RAN translation starts at CUG codon and undergoes frameshifting to produce poly-GP and poly-GR in the +2 and +3 frames, respectively. The double mutation -3/+4 **GCUCUGG>UCUCUGC** in the Kozak sequence

(new Figure 3b,e,f and Supplementary Fig. 1 construct #11) severely reduced the level of poly-GA and prevented the production of poly-GP and poly-GR. These data are corroborated with ^{35}S -Met incorporation experiments (new Figure 3e). Overall, these new experiments confirm that G_4C_2 RAN translation shares similar mechanisms with canonical translation, including a very efficient translation when the start codon is in perfect kozak sequence context. They also confirmed that CUG is indeed the start codon for DPRs production using an initiator Met-tRNA.

5. Figure 4b. It is bothersome that mutating the CUG codon to CCG does not inhibit the production of poly-GP in HEK293 cells, as it does in RRL (Figure 3d). The authors suggest that in cells the poly-GP translation occurs from an alternative start codon and is influenced by trans-acting factors that are absent in RRL. This raises questions about the relevance of the use of the RRL system to study the translational mechanism. Could they rescue poly-GP synthesis from the mutated construct in RRL by adding an extract from HEK293 cells?

We performed the suggested experiment by supplementing RRL system with HEK293 lysates and measuring ^{35}S -Met incorporation and DPR levels. We mainly observed that HEK293 extracts inhibit global translation in our system (Figure 1 inserted below). Indeed, the level of beta-globin is strongly reduced with an increased concentration of HEK293 lysates. Choosing a low concentration of HEK293 lysate that does not affect overall translation did not increase the level of GP. This experiment is not conclusive and we think that HEK293 and RRL extracts might contain other translational activators/inhibitors affecting the specific trans-acting factor effect.

Interestingly, during the revision of this manuscript, DDX21 was identified in HEK293 cells as an RNA helicase able to bind and unwind RNA containing G-quadruplex (McRae et al., 2017). Since G-quadruplex structures are formed by C9ORF72 G₄C₂ repeats, DDX21 is a very attractive candidate as modifier of RAN translation in different systems. We now discuss this new finding in our manuscript. In addition, to further explore whether cell-type specific factors influence RAN translation, we have tested G₄C₂ RAN translation and the impact of CUG mutation in human neural progenitors. RAN translation of poly-GA but also poly-GP were both prevented by mutating the near-cognate codon, recapitulating the results observed in RRL (**new Figure 4b, c**) and confirming that cell type-specific factors intervene in the production of poly-GP in HEK293 cells.

Reference related to this comment:

- McRae EKS, Booy EP, Moya-Torres A, Ezzati P, Stetefeld J, McKenna SA. Human DDX21 binds and unwinds RNA guanine quadruplexes. (2017) *Nucleic Acids Res.* 45:6656-6668.

6. Figure 7. It is concluded that the G₄C₂ containing transcripts sequester ribosomal subunits, and presumably inhibit global translation. This result is rather preliminary. Could they to test this prediction by exploring the effect of these transcripts' sequestration, in trans on the translation of a reporter mRNA in RRL?

As stated in point #3, we used untreated RRL that still contain lipoxygenase and beta-globin mRNA. In all the experiments presented in the manuscript, we do not see a concomitant reduction of the translation of these two mRNAs when G₄C₂ mRNA is translated, suggesting that G₄C₂ RNA does not show a global translation inhibitory effect at 100 and 200 nM. This can be explained by the presence of ribosome in large excess in RRL, that request probably very high amount of G₄C₂ repeats or very long repetitions as observed in patients to induce translation inhibition. However, we agree that our experiments do not demonstrate that the global translation is inhibited in C9ORF72 patient cells and we have toned down this statement.

Notably, we now have more evidence that ribosomal sequestration is dependent on G₄C₂ repeats, as the antisense C₄G₂ repeat transcripts, that undergoes RAN translation in C9ORF72 patients, do not sequester ribosomes (**new Figure 7a-c**). Importantly, we have also included an additional control with G₄C₂ transcripts alone on sucrose gradients (**new Figure 7b**). Indeed, repeat expansion-containing RNAs were recently shown to undergo abnormal phase transition leading to the formation of gel-like structures in vitro (Jain and Vale, 2017). With this new experiment (**Figure 7b**), and the use of purified ribosomal subunits (**Figure 7e**), we demonstrate that migration of G₄C₂ RNAs to the heavy fractions of sucrose gradient is due to sequestration of ribosomal subunits rather than a phase transition phenomenon.

Reference related to this point

- Jain A, Vale RD. RNA phase transitions in repeat expansion disorders. (2017) *Nature* 546:243-247.

7. Figure 7e shows the position of RNP. This makes no sense as their assays were done only with mRNA and ribosomal subunits. Do I miss something?

The reviewer is right and we have now replaced “RNP” by “free RNA” on Figure 7e.

Minor comments:

1. Page 3 "...process that requires numerous elongation initiation factors (eIFs)." should read "...process that requires numerous eukaryotic initiation factors (eIFs)."

The correction has been made.

Reviewer #2 (Remarks to the Author):

This manuscript examines the ALS/FTD C9orf72 gene and its expression as RAN translation. From the studies conducted, the authors conclude that RAN translation of this mRNA occurs in a cap-dependent manner utilizing a CUG codon for initiation. They also find that these transcripts are sticky/bind to either 40S or 60S subunits.

Major concerns

1. The authors use the IGR from the Cricket paralysis virus as a control for "efficient translation". However, it would seem more appropriate to use a normal, cap-dependent reporter to see just how efficiently the repeat transcripts are expressed (i.e. a globin mRNA derivative). Secondly, it is curious that there appears to be little dependency on mRNA input with only the GR product showing an increase with increased RNA (66) although there is a decrease with added RNA for the 30 repeats. Third, as relates to the relative synthesis of either GA, GP or GR, is it possible that this reflects the tRNA populations present in RRL? An examination of the rabbit beta globin chain mRNA indicates the following use of codons that might arise from the G₄C₂ repeat: arginine – no CGG codons used; alanine – half of the codons used are GCC; proline – no CCG codons used; glycine – more than half of the codons used are either GGG or GGC. If one assumes that in the reticulocytes, which are synthesizing 95-98% hemoglobin, that the tRNA population is a match to the amino acids in hemoglobin, might this then be reflected in the synthesis seen in Figure 1, panels A, B and C?

We have followed the reviewer recommendation and compared RAN translation to the translation of both a cap-dependent reporter and an IRES driven reporter. Indeed, we have compared the translation efficiency of the Renilla Luciferase gene under the control of either the beta-globin 5'UTR or the IGR (**new Supplementary Fig. 2**). Efficiency was measured by ³⁵S-methionine incorporation and luminescence. Both showed that translation driven by the beta-globin 5'UTR is two times more efficient than IGR. We showed in Fig. 1 of the manuscript that poly-GA RAN translation is eighteen times more efficient than IGR, hence approximately 9 times more efficient than the beta-globin 5'UTR. Poly-GP and poly-GR translation is equivalent to IGR, and twice less efficient than beta-globin. Overall, G₄C₂ RAN translation is a very efficient mechanism, considering that beta-globin is highly translated in RRL.

Concerning the RNA concentration dependency, DPR level is increased with the concentration of transcripts containing 66 repeats. Translation efficiency of poly-GA is very high when transcripts are capped and reach saturation, but when the efficiency is lower we can observe that poly-GA levels increase with RNA concentration (**Figure 1a**; uncapped versus capped 66 repeats). We agree that the translation efficiency of poly-GR with 30 repeats is less sensitive to the RNA concentration. This experiment was repeated several times and

we do not have an explanation so far about why increasing RNA concentration is inhibitory to GR expression.

Finally, concerning the tRNA concentration from rabbit reticulocyte lysates, we agree that translation in RRL could be influenced by globin expression and be responsible for different rates of expression. However, it is noteworthy that the RRL used in our study are self-made extracts, supplemented with total tRNAs purified from rabbit liver to activate the system and therefore erasing any potential reticulocyte-specific tRNA content adaptation. In addition, the general codon usage of *Oryctolagus cuniculus* indicates that the codons used for DPR synthesis (highlighted in yellow) are not considered as rare codons except for CCG (Pro) (see below, from <http://www.kazusa.or.jp>). RRL also contain lipoxygenase mRNA and this mRNA contains 4 CCG codons. The fact that we don't see any trans-inhibitory effect of G₄C₂ mRNA on both globin and lipoxygenase synthesis (see response to point 3 of Reviewer 1) demonstrates that the corresponding tRNAs are indeed not limiting.

Oryctolagus cuniculus [gbmam]: 1115 CDS's (529901 codons)											
fields: [triplet] [frequency: per thousand] ([number])											
Pro											
UUU 16.4(8668)	UCU 10.4(5510)	UAU 10.0(5281)	UGU 8.2(4365)	UUC 28.4(15072)	UCC 19.4(10268)	UAC 20.0(10614)	UGC 13.6(7188)	UUA 5.3(2822)	UCA 7.7(4062)	UAA 0.6(313)	UGA 1.1(582)
UUG 10.9(5800)	UCG 5.7(3013)	UAG 0.5(279)	UGG 14.1(7451)	CUU 10.1(5332)	CCU 12.6(6683)	CAU 7.3(3889)	CGU 3.7(1955)	CUC 23.6(12503)	CCC 20.8(11041)	CAC 16.0(8496)	CGC 13.0(6905)
CUA 4.9(2604)	CCA 11.8(6265)	CAA 9.2(4889)	CGA 5.0(2668)	CUG 48.9(25898)	CCG 8.7(4595)	CAG 33.0(17503)	CGG 11.4(6056)	AUU 14.3(7594)	ACU 9.9(5239)	AAU 13.4(7119)	AGU 8.5(4526)
AUC 29.7(15747)	ACC 22.0(11660)	AAC 24.2(12813)	AGC 19.3(10232)	AUA 6.1(3233)	ACA 11.7(6174)	AAA 20.2(10709)	AGA 9.2(4872)	AUG 24.3(12876)	ACG 9.1(4800)	AAG 35.1(18610)	AGG 10.6(5599)
GUU 8.7(4602)	GCU 15.5(8220)	GAU 17.6(9311)	GGU 8.8(4671)	GUC 18.0(9528)	GCC 34.2(18137)	GAC 30.5(16155)	GGC 26.7(14143)	GUA 4.8(2567)	GCA 12.7(6710)	GAA 24.2(12830)	GGA 14.7(7798)
GUG 33.3(17648)	GCG 9.5(5045)	GAG 43.7(23167)	GGG 17.0(8996)								
Coding GC 54.73% 1st letter GC 56.01% 2nd letter GC 40.65% 3rd letter GC 67.53%											
Ala											

2. A more convincing proof that RAN translation initiates with methionine would be to add poly(IC) to activate PKR and show that phosphorylation of eIF2 reduces expression of the peptides. It is noted even for globin synthesis that the N-terminal, initiating methionine is removed and thus the only methionine registered is from an internal methionine. Based upon the N-end rule, the amino acid coded for following the CUG codon would be glutamic acid (GAA) and this should result in the removal of the N-terminal methionine (see Huang et al. Biochemistry 1987).

We agree with the reviewer that the N-terminal methionine is usually processed. However, using the histone H4 mRNA (which contains the initiator methionine and a single internal methionine), we have previously determined that in our self-made Rabbit Reticulocyte lysates, only ~60-70% of the N-terminal methionine is processed leaving a residual N-terminal ³⁵S-methionine (Martin et al., 2011; Martin et al., 2016). An incomplete processing of the N-terminal methionine was also found *in vivo* with ~20% of the N-terminal methionine being acetylated instead of removed (Gigliante et al., 2015). In this manuscript, none of the constructs, #3 to #11, used to study the G₄C₂ translation contain any AUG codon in none of

the three frames (**Supplementary Fig. 1; Table S1**). Hence, the DPR products do not incorporate any internal methionine and the radiolabelled DPR products that we observe at the expected size (**Figure 2**) derive from the incorporation of N-terminal ³⁵S-methionine. This has now been clarified in the text and we have included a control without G₄C₂ repeats to confirm that the ³⁵S-labelled product is dependent on the translation of the G₄C₂ repeat transcripts (**new Figure 2b**). Consistently, ³⁵S-labelled peptides are immunoprecipitated by an antibody against the HA tag which is in frame with the poly-GA dipeptide repeat proteins (**Figure 2c**).

In addition, it is increasingly recognized that processing of the N-terminus is influenced by the nature of the second amino-acid (Frottin et al., 2006; Martinez et al., 2008). We used the “Terminator” software available on line (<https://bioweb.i2bc.paris-saclay.fr/terminator3/>) to predict the N-terminus of mature poly-GA and control renilla luciferase proteins. The poly-GA N-terminus is predicted with a likelihood of 100% to be a methionine, meaning unprocessed, due to the presence of a glutamic acid residue in the second position, while the methionine of the renilla luciferase N-terminus is predicted to be processed with a likelihood of 77% (see Table 1 in response to Reviewer 1).

As suggested, we inhibited the ternary complex formation by inducing the phosphorylation of eIF2 alpha subunit (poly (I:C)/salubrinal treatment) thereby inhibiting the recruitment of methionylated initiator tRNA. Such a treatment inhibits ³⁵S-methionine incorporation of canonical cap-dependent translation but not IGR-driven translation which does not use the initiator tRNA to start translation. This treatment also inhibits ³⁵S-Methionine incorporation in DPR products, confirming the methionine incorporation by methionylated initiator tRNA and the involvement of eIF2 (**new Supplementary Fig. 4**).

Along with other experiments described in the answer to Reviewer 1 (comment #1), these results demonstrate that RAN translation of G₄C₂ is indeed initiated by a methionine residue.

3. The authors do show that the translation of their transcripts is favored when the mRNA is capped. However, what is the evidence that *in vivo* the mRNA responsible for RAN translation is capped?

The cap is recognized by the complex eIF4F (eIF4E, eIF4G and eIF4A) allowing the recruitment of the 40S ribosomal subunit at the 5' end of the mRNA. We showed in RRL that treatment with FL3, a specific inhibitor of the RNA helicase eIF4A, inhibits G₄C₂ RAN translation (Figure 6b and Supplementary Fig. 6b,c). We now have tested the impact of FL3 treatment on RAN translation *in vivo* by treating HEK293 cells transfected with construct #4. Translation of poly-GA, GP and GR RAN was severely reduced supporting the importance of eIF4F and a cap-dependent scanning mechanism for RAN translation in human cells (**new Figure 6c-e**).

4. What is the evidence that the CUG initiating codon does in fact direct the binding and use of Met-tRNAⁱ (see above concern)? The use of CUG codons and leucyl-tRNA have shown up in several recent publications.

Please, see our response to Reviewer 1 (point 1) and Reviewer 2 (point 2).

5. Figure 7 – binding of GGGGCC transcripts to ribosomal subunits. This experiment is uncontrolled. It would appear that the transcripts are being bound non-specifically. Controls such as globin mRNA or the IRG segment used in Figure 1 should be used to ensure that the

observed binding is of some relevance. This is especially worrisome for the apparent “polysome-like” aggregates seen in Panel E.

We do agree with the reviewer that it was unexpected to observe that G_4C_2 expanded transcripts can bind ribosomes independently from RAN translation. We have now included several controls to reinforce this result. First, we migrated free G_4C_2 repeat RNAs on sucrose gradient without any extracts or factors to ensure that the RNA itself was not undergoing gel formation mimicking a polysome profile. Indeed, repeat expansion-containing RNAs were recently shown to undergo abnormal phase transition leading to the formation of gel-like structures in vitro (Jain and Vale, 2017). G_4C_2 RNAs do not sediment with the heavy fractions but remain in the light fractions (**new Figure 6b**). Second, we determined that, contrary to the sense G_4C_2 transcripts, the antisense C_4G_2 transcripts with 66 repeats migrates in the light fractions on sucrose gradient (**new Figure 7b**). Treating the RRL extract with edeine prevents ribosomes association for antisense but not sense transcripts, confirming the binding specificity of the G_4C_2 containing RNAs (**new Figure 7c**). We also show that ribosomes assembly with capped beta-globin transcripts, but not G_4C_2 repeats, is affected by cycloheximide, edeine and GMP-PNP (**new Figure 7a-d** and **Supplementary Fig. 6e-h**). Finally, none of the constructs tested in our laboratory other than G_4C_2 transcripts, including the histone H4, showed a “polysome like profile” when using purified ribosomal subunits (**Figure 7e**).

Reference related to this comment:

- Jain A and Vale RD. RNA phase transitions in repeat expansion disorders. (2017) Nature 546:243-247.
- Martin F, Barends S, Jaeger S, Schaeffer L, Prongidi-Fix L, Eriani G. Cap-assisted internal initiation of translation of histone

Minor concerns

1. The authors would benefit from reviewing their manuscript for better use of English and to remove some technical errors (i.e. Introduction – “... that requires numerous elongation initiation factors (eIFs)...” The e in eIF stands for eukaryotic, not elongation.).

This error has been corrected.

2. Why is the level of cap inhibitor (m7GpppG) so high to affect inhibition (1.5 mM). Often in other studies, the level used was in the 100 micromolar range. This level represents a 15,000 to 1 ratio of analog to mRNA.

As shown in Figure 2d, the principle of this competition assay is to saturate the endogenous eIF4E factor by blocking its cap-binding pocket with a cap analog. The whole pool of eIF4E has to be neutralized, hence the inhibiting concentration of cap analog is not related to the mRNA concentration but rather to the amount of eIF4E present in the RRL. In addition, an excess of cap analog is required to prevent its dissociation from eIF4E. To support our initial finding, we have performed the same competition assay using Wheat Germ Extracts that are fully cap-dependent (**new Supplementary Fig. 4c,d**). The results are comparable in Wheat Germ Extracts and RRL supporting a 5' scanning mechanism for RAN translation of G_4C_2 transcripts.

3. Figure 2 – it is not clear from the transcripts whether there are any internal methionines in the coding region. This should be checked and reported.

We apologize for the confusion and have now clarified in the manuscript that “the sequence of the transcripts #3 and #4 do not contain any AUG codon and the presence of [³⁵S]-methionine in RAN products cannot derive from the incorporation of an internal methionine (**Supplementary Figure 1, Table 1**)”.

Reviewers' comments:

Reviewer #1 (Remarks to the Author):

The revised manuscript by Tabet and co-authors and their point by point response satisfactorily addresses my major concerns.

They now show that mutating of the CUG codon to AUG in the GA reading frame significantly enhances expression of GA-NLuc while decreasing expression in the GP frame both in vitro and in vivo. This is consistent with competition for initiation between GA and GP reading frames. However, evidence for competition between GA and GR reading frame is less clear given the discrepant results with HEK293 cells and RRL (new Supplementary Fig. 2d, f).

I agree with their reasoning that less efficient binding of eIF4F to cap analogs as compared to capped mRNAs could be responsible for the relatively small inhibition of the control AUG mRNA by the m⁷GpppG cap analogue.

An intriguing observation is that the loss of the upstream near-cognate CUG start codon significantly stimulates RAN translation in the GP reading frame in RRL. The authors speculate that the scanning PIC can initiate translation within the repeat itself, and that the removal of CUG increases the fraction of PICs reaching the repeat. Would it be possible to provide any experimental support for such a mechanism?

Reviewer #2 (Remarks to the Author):

This is an improved manuscript that has addressed most, but not all of previous concerns. The positive feature is the identification of a non-conical start codon CUG. The unique feature is the analysis that suggests that this codon is the only start codon and that only one of the three dipeptide repeats is in frame with this codon. The required frame shift (proposed) for the other two reading frames has not been seen by others.

Concerns

1. Figure 2 – The ratio of globin synthesis to the peptide repeats would suggest that the dipeptide repeats are going via a minor route for initiation and one that is less efficient. This may also be evident for the proteins synthesized as marked by an asterisk (*). Previously a protein of 41,000 Da was found to be radiolabelled using RRL and this was independent of either added mRNA or added puromycin. This was called the “Kaji system” and represented the addition of methionine to the N-terminus of an existing protein. What is the absolute quantitation of synthesized globin to the dipeptide repeat? And since the synthesis of the GA dipeptide is about 18 times greater than that of the others, what is the chance that these may represent a different initiation mechanism? Additionally, why is lipoxxygenase synthesis inhibited by the 30 repeats but not by the 60 repeats in panel b?
2. Figure 3 – why is there a decrease in lane 7 of panel c or is this just variable results day to day? Second, is there anything unusual about construct 8 which has only 9 nucleotides upstream of the initiating CUG codon?
3. What is the quantitative reduction in globin synthesis in the presence of poly(I:C)-(in Figure S4b)? By eye, the reduction appears to be quite small, perhaps indicating that a non-optimal level of poly(I:C) has been added.
4. The results seen in Figure 4d are inconsistent with a frame shifting event causing the synthesis of the GP dipeptide repeat.
5. It is likely that the results in Figure 6b are only valid if there is an equal reduction in total protein synthesis with each of the inhibitors. It would appear that the level of global inhibition of protein

synthesis under the conditions used is greater with CHX or Edeine.

6. The results in Figure 7 appear inconsistent. In panels d and e, why is there such a spread with the (G4C2)₃₀ RNA? Why are there polysomes with the (G4C2)₆₆ RNA when there is none with beta-globin mRNA? How many (G4C2) repeats does it take to obtain polysomes? Is the binding of the repeats sensitive to salt (it is noted that 50 mM is rather low salt)?

7. There is no description as to how or why frame shifting might occur. Second, a brief examination of the sequence in the coding region between the CUG start and the repeat does not provide evidence for a slippery sequence (i.e. a homopolymeric stretch, especially A's or U's). Second, frameshifting is usually a very rare event occurring at most at the 1% level unless a slippery sequence is present (i.e. RF2 can frameshift with a frequency approaching 50%). Thus, it is unclear to this reviewer that frameshifting is the cause for the synthesis of the GP and GR dipeptide repeats.

8. This reviewer is not convinced that the loading of ribosomal subunits on the nucleotide repeat represents a significant biological finding (and is more likely an artifact). Proof of functionality would require a correlation of extent of nucleotide repeat binding with increased protein synthesis of the dipeptide repeats.

Revision of the manuscript “ALS/FTD C9ORF72 transcripts initiate translation at a CUG codon and sequester ribosomal subunits” (NCOMMS-17-10398B) submitted by Tabet et al.

Reviewer #2 (Remarks to the Author):

This is an improved manuscript that has addressed most, but not all of previous concerns. The positive feature is the identification of a non-conical start codon CUG. The unique feature is the analysis that suggests that this codon is the only start codon and that only one of the three dipeptide repeats is in frame with this codon. The required frame shift (proposed) for the other two reading frames has not been seen by others.

Concerns

1. Figure 2 – The ratio of globin synthesis to the peptide repeats would suggest that the dipeptide repeats are going via a minor route for initiation and one that is less efficient. This may also be evident for the proteins synthesized as marked by an asterisk (*). Previously a protein of 41,000 Da was found to be radiolabelled using RRL and this was independent of either added mRNA or added puromycin. This was called the “Kaji system” and represented the addition of methionine to the N-terminus of an existing protein. What is the absolute quantitation of synthesized globin to the dipeptide repeat? And since the synthesis of the GA dipeptide is about 18 times greater than that of the others, what is the chance that these may represent a different initiation mechanism? Additionally, why is lipoxxygenase synthesis inhibited by the 30 repeats but not by the 60 repeats in panel b?

We have chosen to use self-made untreated RRL because it has been previously described that RNase treatment (used in commercial available extracts) is detrimental to the efficiency of the translation, especially in terms of cap-dependency (Soto Rifo et al., 2007). In this system, the globin is translated from a large pool of existing endogenous mRNA encoding β -globin. However, we are using sub-saturating G_4C_2 mRNA concentrations for DPR synthesis to avoid titration effects (**Fig 2b**, the DPR synthesis is increasing with higher G_4C_2 RNA concentrations), which is not the case for the β -globin mRNA present in the lysates. In addition, it is well established that globin translation is extremely high in RRL (Nienhuis and Benz N, 1977; Mills et al., 2017). Lastly, globin and lipoxxygenase contain two and seventeen methionine respectively, while G_4C_2 mRNA does not encode any methionine and ^{35}S -Met radiolabelled DPR products contain only the methionine incorporated at the CUG start codon. In summary, RNA concentrations and number of ^{35}S -methionine incorporated in the proteins are different between DPR, β -globin and lipoxxygenase. Therefore, we quantitatively compared DPR to IRES-dependent translation, using the same tags for immunoblot experiments and the same equimolar RNA concentrations (**Fig. 1**). Then, following the reviewer recommendations, we used the renilla luciferase coding sequence as reporter to compare the level of capped β -globin 5'UTR to IRES-dependent translation and determine the IRES/cap-dependent translation ratio in our system (**Fig S2**). Translation triggered by the 5'UTR of the β -globin is two times higher than the IRES-dependent translation. Thus, the efficiency of poly-GA translation is 16 times more than IRES-mediated translation compared with HA tag (**Fig. 1**), and 8 times more than translation triggered by capped β -globin 5'UTR.

We also would like to stress that G_4C_2 repeats RNA constructs do not harbor any AUG codon encoding methionine. Thus, incorporation of ^{35}S -methionine is only due to the N-terminal incorporation of a methionine in DPR products. We provide several pieces of evidence demonstrating that addition of methionine at the N-terminus of a pre-existing DPR protein is not occurring in our system and that ^{35}S -Met products migrating at the expected size

correspond to the translation of either 30 or 66 G₄C₂ repeats. Indeed, a single point mutation CUG>CCG, as well as mutations removing the CUG codon, abolish the incorporation of ³⁵S-Methionine (**Fig. 3c**). In addition, ³⁵SMet-DPRs are successfully immunoprecipitated with an antibody against HA-tag in frame with poly-GA (**Fig. 2c**). Furthermore, ³⁵SMet-DPRs translation is enhanced when the CUG codon is mutated to a canonical AUG, and decreased when the Kozak sequence is mutated (**Fig 3e**). Finally, ³⁵S-radiolabelled DPR is inhibited by an existing uORF (**Fig 5c**), ASOs targeting the 5' flanking sequence (**Fig. 6f,g**) and translation inhibitors targeting eIF2-Met-tRNA^{Met} complex (**Supplementary Fig. 4**).

We apologize for the confusion but Lipoxigenase is not inhibited in Figure 2b. Indeed, the band on top of the gel that was marked by an asterisk actually corresponded to the stacking of the gel. We thank the reviewer for pointing this error and have now replaced the figure to show the entire gel and correctly annotate the band corresponding to Lipoxigenase.

References:

- Soto Rifo et al. Back to basics: the untreated rabbit reticulocyte lysate as a competitive system to recapitulate cap/poly(A) synergy and the selective advantage of IRES-driven translation. (2007) Nucl. Acids Res 35, e121.
- Nienhuis AW and Benz,EJ. Regulation of Hemoglobin Synthesis during the Development of the Red Cell (1977) N Engl J Med, 297 :1318-1328.
- Mills EW, Green R and Ingolia N. Slowed decay of mRNAs enhances platelet specific translation (2017) Blood 129 :e38-e48.

2. Figure 3 – why is there a decrease in lane 7 of panel c or is this just variable results day to day? Second, is there anything unusual about construct 8 which has only 9 nucleotides upstream of the initiating CUG codon?

We agree with the reviewer that RAN translation is moderately decreased in lane 7 in **Figure 3c** corresponding to the construct #7. These results are reproducible (the gel provided in **Figure 3** is representative of results obtained from 3 independent experiments), and consistent with the immunoblot results in **panel d** of the same figure, with a moderate decrease of the translation in all reading frames. We have now clearly stated this result in the text.

A slight reduction of RAN translation is also observed with construct #8, harboring only 9 nucleotides upstream of the CUG near cognate start codon (**Fig 3c,d**). As noted by the reviewer, the efficiency of the small subunit scanning is expected to be affected by short 5' UTR sequence (Kozak 1991; Martin et al., 2011; Elfakess et al., 2011). Indeed, the ribosome ideally covers 10 to 15 nucleotides upstream of the start codon. Nevertheless, we have previously shown that translation can still be efficient with a 9nt-5'UTR when a stable structure is located downstream of the start codon (Martin et al., 2011; Martin et al., 2016). Consistently, construct 8 is still efficiently translated in all three frames despite the short 5'UTR sequence (**Fig 3c,d**).

References:

- Kozak M. A short leader sequence impairs the fidelity of initiation by eukaryotic ribosomes. (1991) Gene Expr. 1, 111–115.
- Martin F, Barends S, Jaeger S, Schaeffer L, Prongidi-Fix Land Eriani G. Cap-Assisted Internal Initiation of Translation of Histone H4 (2011) Mol Cell 41, 197-209.
- Elfakess R, Sinvani H, Haimov O, Svitkin Y, Sonenberg N, Dikstein R. Unique translation initiation of mRNAs-containing TISU element. (2011) Nucleic Acids Res. 39:7598-609.

Martin F, Menetret JF, Simonetti A, Myasnikov AG, Vicens Q, Prongidi-Fix L, Natchiar SK, Klaholz BP, Eriani G. Ribosomal 18S rRNA base pairs with mRNA during eukaryotic translation initiation. (2016) Nat Commun Aug 24;7:12622.

3. What is the quantitative reduction in globin synthesis in the presence of poly(I:C)-(in Figure S4b)? By eye, the reduction appears to be quite small, perhaps indicating that a non-optimal level of poly(I:C) has been added.

For the experiment in Figure S4b, we have used 150 ng/mL of poly(I:C) and 15mM of salubrinal as previously described by Namer et al., Cell reports (2017). This reference and a more detailed description of the method have now been included in the manuscript. Reviewer 2 suggested that the poly(I:C) concentration used in this assay may be too low. In order to address this point, we repeated this experiment using twice more poly(I:C) (300 ng/mL) and salubrinal (30mM) and obtained the same inhibition of cap-dependent translation as in Figure S4 (Figure below). This experiment, along with the results mentioned in comment #1, demonstrates that DPR synthesis is using the methionylated initiator tRNA^{Met}.

Reference:

Namer et al. An ancient pseudoknot in TNF-alpha pre-mRNA activates PKR, inducing eIF2alpha phosphorylation that potently enhances splicing. (2017) Cell Reports 20, 188-200.

4. The results seen in Figure 4d are inconsistent with a frame-shifting event causing the synthesis of the GP dipeptide repeat.

The approach we have taken has the strength to allow following RAN translation in all reading frames from a unique G₄C₂ RNA construct. Indeed, using different constructs for each frame would not permit the identification of frameshifting events. We provide several pieces of evidence supporting that a single CUG start codon is used for translation of the three frames with frameshifting events observed in vitro and in cellular models. Importantly, we now show that poly-GP translation is strongly inhibited by mutation of the CUG codon in reticulocyte lysates (RRL), and in 2 neuronal models (human neuronal progenitors and mouse motor neuron like cells NSC-34). However, we also observe that cell-type specific factors influence RAN translation in the poly-GP frame that is differently translated in HEK293T cells than in the 3 other models. In summary we have found that:

1- Mutation of CUG>CCG abolishes DPR translation from all 3 reading frames in reticulocyte lysates, supporting a frameshifting mechanism to produce poly-GP and poly-GR (Fig 3d).

2- Mutation of the near cognate CUG codon into a canonical AUG start codon (CUG>AUG) enhances expression of the three DPRs poly-GA, poly-GP and poly-GR in reticulocyte lysates (Fig 3d,f, compare construct #10 to #4). Furthermore, mutations of the CUG Kozak sequence GCUCUGG>UCUCUGC inhibit equally DPR expression from the 3 frames, confirming that

translation of all DPRs is initiated from the same start codon (**Fig. 3e,f**, compare construct #11 to #4).

3- Mutation of the near-cognate start codon CUG>CCG in human neural progenitors (**Fig. 4b,c**) and mouse motor neuron-like cells NSC34 (**new Fig. 4d,e**) also abolishes the translation of both poly-GA and poly-GP, validating a frameshifting mechanism to produce poly-GP in human neuronal progenitor cells (**Fig. 4b,c**). Poly-GR levels were low in all models and undetectable in neuronal cells preventing conclusion about poly-GR in these models.

4- The same mutation CUG>CCG abolishes both poly-GA and poly-GR translations in kidney cells (HEK293T) (**Fig. 4f-g**), confirming a frameshifting mechanism to produce poly-GR.

5- However, contrary to RRL, motor neurons NSC-34 and human neural cells, poly-GP level is increased by CUG>CCG mutation in HEK293T cells, demonstrating that cell-type specific factor(s) are influencing poly-GP expression in absence of CUG codon in HEK293T cells. Notably, there is a stop codon at the beginning of G₄C₂ repeats (UAGGGGCC) in frame with poly-GP. Thus, frameshifting before this stop codon is necessary to translate poly-GP, unless the translation occurs in the repeats itself. DDX21 was recently shown to bind and unwind RNA guanine quadruplexes in HEK293T (McRae et al., 2017) and represents a good candidate factor to influence the G-quadruplex structures and increase poly-GP production in absence of CUG codon in HEK293T.

These results have been obtained from more than 3 independent experiments for each model and we believe that it is important to report the divergent regulation of RAN translation in various cellular models.

5. It is likely that the results in Figure 6b are only valid if there is an equal reduction in total protein synthesis with each of the inhibitors. It would appear that the level of global inhibition of protein synthesis under the conditions used is greater with CHX or Edeine.

CHX and edeine are translation inhibitors that bind specifically the ribosome (Garreau de Loubresse et al., 2014), blocking directly the translocation step and the interaction between the Met-tRNA^{Met} anticodon and the initiator codon in the P-site of the ribosome. In contrast, FL3 and 4EIRCAt are translation inhibitors that affect initiation factors and the scanning step, indirectly blocking the ribosomal assembly. Hence, the effect of CHX and edeine is more robust than the translation inhibition induced by FL3 and 4EIRCAt, two inhibitors of initiation factors. In addition, mRNAs are differentially sensitive to scanning inhibitors depending on their 5'UTR length and the secondary structures that are encountered during scanning (Iwasaki et al., 2016 ; Pelletier et al., 2015). This different sensitivity to scanning inhibitors is illustrated in **Figure S6a,c** where we observed a more drastic translation inhibition by FL3 when using construct #2 that contains a longer 5'UTR than construct #4.

References :

Garreau de Loubresse et al. Structural basis for the inhibition of the eukaryotic ribosome. (2014) Nature 513, 517-522.

Iwasaki S, Floor SN and Ingolia NT. Rocaglates convert DEAD-box protein eIF4A into a sequence-selective translational repressor. (2016) Nature, 534, 558-561.

Pelletier J, Graff J, Ruggero D and Sonenberg N. Targeting the eIF4F Translation Initiation Complex: A Critical Nexus for Cancer Development. (2015) Cancer Res 75, 250-263.

6. The results in Figure 7 appear inconsistent. In panels d and e, why is there such a spread with the (G4C2)₃₀ RNA? Why are there polysomes with the (G4C2)₆₆ RNA when there is

none with beta-globin mRNA? How many (G₄C₂) repeats does it take to obtain polysomes? Is the binding of the repeats sensitive to salt (it is noted that 50 mM is rather low salt)?

To analyze the ribosome assembly on each mRNA (**Fig 7a**), we radiolabelled them at their 5' end by capping. The radioactive mRNAs were incubated in cell-free translation extracts or with purified ribosomal subunits and then loaded onto sucrose gradients for sedimentation. After gradient collection, we monitored the position of the mRNA by measuring the radioactivity of all the sucrose gradient fractions. The profiles presented in **Figure 7** and **Supplementary Figure 6e-i**, represent the radioactive counts, which correspond to the radioactive mRNA throughout the whole gradients.

The major finding of this experiment is the abnormal sedimentation of both (G₄C₂)₃₀ and (G₄C₂)₆₆ RNAs into heavy polyribosome fractions even when translation was blocked by different inhibitors. Translation was efficiently inhibited by addition of CHX 5 minutes prior to the addition of radiolabelled RNAs to RRL, as demonstrated by the migration of control β -globin RNA in light fractions with a high peak corresponding to the monosome fraction (**Fig. 7d**). Similar observations were obtained with other inhibitors such as edeine that blocks 43S complex (**Fig. 7c**) or GMP-PNP blocking the assembly of the 60S subunit (**Supplementary Fig. 6g**).

Additional evidence supporting the sequestration of ribosomal subunits onto G₄C₂ RNA independently of translation is the observation that G₄C₂ RNAs also migrate in heavy fractions when incubated with purified 40S or 60S ribosome subunits (**Fig 7e**).

We provide several controls in this experiment, including the demonstration that antisense (C₄G₂)₆₆ RNAs do not sequester ribosomal subunits (**Fig 7b,c**). In addition, G₄C₂ repeat RNAs were recently shown to undergo liquid droplet formation. We verified that G₄C₂ RNA itself without RRL is not found in the heavy fractions, demonstrating that G₄C₂ RNAs are migrating in heavy fractions only in presence of ribosomal components. Overall, We provide evidence that ribosomal components sequestration is independent from RAN translation.

The reviewer is right that salt concentration may influence RNA structure and association with ribosomal subunits of RNAs. Following his/her suggestion we have performed a new experiment showing that migration of G₄C₂ transcripts in the heavy fraction is further increased when they are folded in presence K⁺ ions that stabilize G-quadruplex structures, comparatively to Na⁺ and Li⁺ ions (**new Supplementary Fig. 6i**; 195mM).

7. There is no description as to how or why frame shifting might occur. Second, a brief examination of the sequence in the coding region between the CUG start and the repeat does not provide evidence for a slippery sequence (i.e. a homopolymeric stretch, especially A's or U's). Second, frameshifting is usually a very rare event occurring at most at the 1% level unless a slippery sequence is present (i.e. RF2 can frameshift with a frequency approaching 50%). Thus, it is unclear to this reviewer that frameshifting is the cause for the synthesis of the GP and GR dipeptide repeats.

There is compelling evidence from the literature that G-quadruplex structures induce translation frameshifting (Endoh and Sugimoto, 2013 ; Yu et al., 2014 ; reviewed in Kapur et al., 2017). This phenomenon can occur with only 1 G-quadruplex structure and is increased by stabilizing the G-quadruplex with specific molecules. We agree with the reviewer that there is no canonical slippery sequence, however the G-quadruplex structures are repeated 16 times in the 66 G₄C₂ constructs and we provide several pieces of evidence supporting frameshifting events for Ran translation of G₄C₂ (please see comment #4).

References :

Endoh T, Sugimoto N. Unusual -1 ribosomal frameshift caused by stable RNA G-quadruplex in open reading frame. (2013) *Analytical chemistry* 85, 11435-11439.

Yu CH, Teulade-Fichou MP, Olsthoorn RC. Stimulation of ribosomal frameshifting by RNA G-quadruplex structures. (2014) *Nucleic Acids Res* 42, 1887-1892.

Kapur M, Monaghan CE, Ackerman SL. Regulation of mRNA Translation in Neurons-A Matter of Life and Death. *Neuron* 96, 616-637 (2017).

8. This reviewer is not convinced that the loading of ribosomal subunits on the nucleotide repeat represents a significant biological finding (and is more likely an artifact). Proof of functionality would require a correlation of extent of nucleotide repeat binding with increased protein synthesis of the dipeptide repeats.

We agree that the results obtained on sucrose gradients are complex and unexpected, but we believe that these data are innovative and validated by the use of multiple controls. Importantly, the ribosome sequestration that we observed is not linked to DPR synthesis. Indeed, ribosomal subunits are sequestered by G₄C₂ repeats in a translational-independent manner since treatment with translation inhibitors did not prevent the migration of repeat-containing transcripts in the heavy fractions (**Fig 7c,d** and **Supplementary Fig 6f-h**). In addition, we observed that G₄C₂ transcripts also sequester purified ribosomal 40S or 60S subunits in absence of translation (**Fig 7e**). On the contrary, β-globin and C₄G₂ antisense RNAs do not sequester ribosomes providing evidence for a specific property of the sense strand G₄C₂ transcripts (**Fig. 7** and **Fig. S6**). We believe that these data represent a significant biological finding, considering that different molecular mechanisms have been proposed in C9ORF72-ALS/FTD, among which the sequestration of RNA binding proteins by expanded RNA foci in the nuclei (Ling et al. 2013). It will be important to further study the relationship between ribosomal proteins and expanded repeats in patient cells, and our observation of G₄C₂ RNAs migrating to heavy fractions on sucrose gradients will open a new research area not yet explored by the field to provide a better understanding of pathogenic mechanisms linked to C9ORF72 expansions.

Reference :

Ling SC, Polymenidou M and Cleveland DW. Converging mechanisms in ALS and FTD: disrupted RNA and protein homeostasis. (2013) *Neuron* 79(3):416-38.

REVIEWERS' COMMENTS:

Reviewer #2 (Remarks to the Author):

This reviewer is satisfied with the responses of the authors and finds that the current manuscript is now acceptable with the following provision: the authors should remove the "sequesters ribosomal subunits" from their title. The rationale for this is that there is no evidence that the repeats inhibit translation (see Figure 1b where the addition of repeat transcripts does not influence the amount of globin made). The authors' possible hint that the repeats sequester (and therefore inhibit) ribosomes has no basis in their experiments. However, it would be worth attempting at some point in time to titrate the repeat transcripts into a translation system to see what concentration of the repeats is necessary to affect inhibition if in fact this "sequestration" does so.

An error on the reviewer's part – in all of the figures where there is the use of two different mRNA levels, there should be a two-fold increase in product formed. In some cases there is (Figures 1c, 2b, 6b) but in some cases there isn't (Figure 1a, 1c (30 repeats), S2c). It is unclear if this represents experimental variation/standard error or if the amount of mRNA added might be out of the linear range. The authors may wish to address this question. This is especially curious where the incorporation of methionine appears to increase in proportion to the added RNA (Figure 2b) but the synthesis of GA does not (Figure 1a).

Final sentence of the results – should probably end in "translational factors" and not "translational actors".

REVIEWERS' COMMENTS:

Reviewer #2 (Remarks to the Author):

This reviewer is satisfied with the responses of the authors and finds that the current manuscript is now acceptable with the following provision: the authors should remove the “sequesters ribosomal subunits” from their title. The rationale for this is that there is no evidence that the repeats inhibit translation (see Figure 1b where the addition of repeat transcripts does not influence the amount of globin made). The authors’ possible hint that the repeats sequester (and therefore inhibit) ribosomes has no basis in their experiments. However, it would be worth attempting at some point in time to titrate the repeat transcripts into a translation system to see what concentration of the repeats is necessary to affect inhibition if in fact this “sequestration” does so.

An error on the reviewer’s part – in all of the figures where there is the use of two different mRNA levels, there should be a two-fold increase in product formed. In some cases there is (Figures 1c, 2b, 6b) but in some cases there isn’t (Figure 1a, 1c (30 repeats), S2c). It is unclear if this represents experimental variation/standard error or if the amount of mRNA added might be out of the linear range. The authors may wish to address this question. This is especially curious where the incorporation of methionine appears to increase in proportion to the added RNA (Figure 2b) but the synthesis of GA does not (Figure 1a).

Final sentence of the results – should probably end in “translational factors” and not “translational actors”.

While we show that G_4C_2 transcripts associate with ribosomal subunits independently of their translation, we agree with the reviewer that we do not have evidence that global translation inhibition is induced through functional sequestration of ribosomes. We have now modified our title and removed the word “sequestration” throughout the text, as well as the sentence “We propose a model where newly synthesized ribosomal subunits are sequestered by G_4C_2 expanded transcripts (Fig. 8d).” from the discussion. The new title is as follows: “CUG initiation and frameshifting enable production of dipeptide repeat proteins from ALS/FTD C9ORF72 transcripts”.

The immunoblot shown in Fig. 1a corresponds to a long exposure allowing detection of RAN translation products from uncapped transcripts on the same immunoblot as capped transcripts and thus does not show the relationship between increased capped RNA concentration and GA production. We have now included a lower exposure of the same immunoblot in Supplementary Fig. 2a to better illustrate the two-fold increase of GA translated from $(G_4C_2)_{66}$ transcripts.

Concerning Supplementary Fig. 2c, we agree that there is not a two-fold increase, however the translational activity from both constructs is increasing in the same range indicating that the experiment was done in sub-saturating conditions. Hence, we concluded that this was appropriate to estimate the relative efficiency of both constructs, which was the goal of this supplementary figure.

The final sentence of the results is now changed to “translational factors”